# Predicting direct and indirect non-target impacts of biocontrol agents using machine-learning approaches

Hannah J. Kotula[1]*, Guadalupe Peralta[1], Carol M. Frost[2], Jacqui H. Todd[3], Jason M. Tylianakis[1,4]

1 Centre for Integrative Ecology, School of Biological Sciences, University of Canterbury, Christchurch, New Zealand, 2 Department of Renewable Resources, University of Alberta, Edmonton, Canada, 3 The New Zealand Institute for Plant and Food Research Limited, Auckland, New Zealand, 4 Bio-Protection Research Centre, School of Biological Sciences, University of Canterbury, Christchurch, New Zealand

* hannahkotula@gmail.com

**Data Availability Statement:** All relevant data are within the manuscript and its Supporting Information files, or on Dryad (https://doi.org/10. 5061/dryad.t5557).

## Abstract

Biological pest control (i.e. 'biocontrol') agents can have direct and indirect non-target impacts, and predicting these effects (especially indirect impacts) remains a central challenge in bio-control risk assessment. The analysis of ecological networks offers a promising approach to understanding the community-wide impacts of biocontrol agents (via direct and indirect inter-actions). Independently, species traits and phylogenies have been shown to successfully pre-dict species interactions and network structure (alleviating the need to collect quantitative interaction data), but whether these approaches can be combined to predict indirect impacts of natural enemies remains untested. Whether predictions of interactions (i.e. direct effects) can be made equally well for generalists vs. specialists, abundant vs. less abundant species, and across different habitat types is also untested for consumer-prey interactions. Here, we used two machine-learning techniques (random forest and k-nearest neighbour; KNN) to test whether we could accurately predict empirically-observed quantitative host-parasitoid net-works using trait and phylogenetic information. Then, we tested whether the accuracy of machine-learning-predicted interactions depended on the generality or abundance of the inter-acting partners, or on the source (habitat type) of the training data. Finally, we used these pre-dicted networks to generate predictions of indirect effects via shared natural enemies (i.e. apparent competition), and tested these predictions against empirically observed indirect effects between hosts. We found that random-forest models predicted host-parasitoid pair-wise interactions (which could be used to predict attack of non-target host species) more suc-cessfully than KNN. This predictive ability depended on the generality of the interacting partners for KNN models, and depended on species' abundances for both random-forest and KNN models, but did not depend on the source (habitat type) of data used to train the models. Further, although our machine-learning informed methods could significantly predict indirect effects, the explanatory power of our machine-learning models for indirect interactions was reasonably low. Combining machine-learning and network approaches provides a starting point for reducing risk in biocontrol introductions, and could be applied more generally to pre-dicting species interactions such as impacts of invasive species.

**Funding:** H.J.K acknowledges support from a Roland Stead Postgraduate Scholarship in Biology. H.J.K, G.P and J.M.T were funded by the Marsden Fund (grant number UOC1705), administered by the Royal Society of New Zealand. J.H.T is funded by the Better Border Biosecurity (B3) (www.b3nz. org) research collaboration funded by the New Zealand Government. We acknowledge that one of the authors (J.H.T) is employed by a commercial company (The New Zealand Institute for Plant and Food Research Limited). Although this company does both government-funded and industry-funded work, there is no industry funding for this research. The funder provided support in the form of salaries for an author [J.H.T], but did not have any additional role in the study design, data collection and analysis, decision to publish, or preparation of the manuscript. The specific role of this author is articulated in the 'author contributions' section.

**Competing interests:** The authors have declared that no competing interests exist. We acknowledge that one of the authors (J.H.T) is employed by a commercial company (The New Zealand Institute for Plant and Food Research Limited). This does not alter our adherence to PLOS ONE policies on sharing data and materials.

## Introduction

Growing public concern about the harmful health and environmental effects of pesticides [1, 2], combined with the rapid evolution of pest resistance to chemical control [3–5], mean that biocontrol agents are increasingly being advocated to suppress pests [6–8]. Although biocontrol agents are often a more environmentally friendly means of suppressing pests than synthetic pesticides, they can have non-target effects on native species [9–11]. Therefore, tools for assessing the risk of proposed biocontrol agents prior to their release are crucial for deciding whether a given agent should be introduced [12, 13]. However, in addition to directly affecting individual non-target species (i.e. direct effects), biocontrol agents have the potential to affect communities through indirect effects (i.e. the effect of one species on another mediated by a third species) [14]. These indirect effects are more difficult to observe (or predict) because of the inherent difficulty in studying whole communities of interacting species and attributing causation to changes in potentially indirectly affected populations. Although various tools (e.g. host range experiments) exist for assessing the potential direct effects of proposed biocontrol agents, to our knowledge, there are currently no methods available for quantifying and ranking potential non-target impacts of proposed biocontrol agents via indirect effects prior to their release [13], despite them being likely to be common [15, 16].

In particular, populations of herbivore species may be linked via their shared natural enemies (e.g. parasitoids, which are often used as biocontrol agents), such that an increase in one host herbivore population may drive an increase in the abundance of a shared enemy and result in a decrease in another host's population (i.e., apparent competition [17]). In this way, parasitoid species may mediate indirect effects, such as apparent competition, among host species. Similarly, herbivores used as control agents of weeds may influence native herbivores via shared enemies. Although typically not quantified in biocontrol risk assessment, many studies indicate that apparent competition, mediated via shared natural enemies, may be important in structuring insect communities [17–21].

Studies assessing apparent competition are mostly focused on its effects between single species pairs, whereas less is known about its potential community-wide impacts. However, quantitative food webs, which include information on the abundance of both consumer and resource species and the frequency of their interactions, can indicate the potential for indirect effects and assist in the generation of testable hypotheses [22, 23]. To test such a hypothesis, a seminal study experimentally removed two leaf-miner species from a community and found that other leaf miners, with which they shared parasitoids, experienced lower parasitism rates [24]. Similarly, Tack *et al.* [25] experimentally increased the abundance of three leaf miner species and, using information on their shared parasitoids, found long-term indirect interactions between some species, as predicted, though these effects were positive (apparent mutualism [17]) rather than negative (apparent competition). These two studies indicate that indirect effects could, in theory, be predicted for all host species in an assemblage. Congruent with this expectation, a recent community-level empirical study found that apparent competition was important for structuring host-parasitoid networks in native and plantation forests, and that these two habitats functioned dynamically (and somewhat predictably) as a single unit [21]. Although the potential negative impacts of apparent competition on native communities have been considered in a biocontrol context [26], it remains difficult to predict these impacts prior to the release of a biocontrol agent.

The analysis of ecological networks is an important tool in theoretical and applied community ecology [27–29], and the above examples suggest that it offers a promising approach to understanding the community-wide impacts (via direct and indirect interactions) of biocontrol agents [11, 30]. This network framework has been widely used in ecology [27–29],

including to quantify and predict indirect effects, and less widely in a biocontrol context to successfully evaluate the impacts of biocontrol agents on non-target species through post-release studies [31–34]. Yet, few studies use it to predict biocontrol outcomes prior to the release of an agent (but see [35]), despite the clear benefits of such an approach [16, 22, 36]. One notable exception was a study by López-Núñez *et al.* [35], which used networks to predict the potential indirect effects of a biocontrol agent. Because networks were only collected prior to the release of the control agent, López-Núñez *et al.* [35] were unable to validate these predictions. Additionally, their predictions of indirect effects were informed by known interaction partners of the control agent (data that may not be available for a given proposed control agent).

One of the limitations to the use of (particularly quantitative) networks in biological control is the cost or effort required to resolve community-scale interactions. A potential solution would be to use frameworks that predict, rather than sample, species interaction networks using existing data on species traits. Species traits play an important role in structuring species interactions and ecological networks [37–40]. For instance, ecological and life history traits determine species distributions, as well as allowing morphological and physiological trait matching between interacting partners, and thereby influence the occurrence and frequency of interactions [38]. When traits cannot be easily measured or the traits responsible for interactions are unknown, phylogenies, which depict the shared evolutionary history of species, can serve as a surrogate for measured and unmeasured traits, and thus help to explain the pattern of interactions within a community [41–44].

Accordingly, many studies have shown that traits and phylogenies can successfully predict direct interactions among species [37, 38, 45, 46]. In addition to traits being useful predictors of interactions, sometimes relatively few traits are needed to explain network structure [45], though in all cases these predictions are of potentially suitable interactions, given the imperfect knowledge of traits and potential for partner co-occurrence to limit the realisation of these potential interactions. More recently, machine-learning techniques, such as random forest and k-nearest neighbour (KNN), have been added to the list of tools used to predict species' interactions based on their traits and/or phylogenies. These algorithms are trained using known interactions, which allow the machine-learning technique to learn which trait and/or phylogenetic combinations are likely to correspond to an interaction occurrence, and these can then be used to predict interactions among new species [47, 48]. In addition to traits and phylogenies (i.e., information that captures niche processes), neutral processes, which describe interaction patterns based on the probability of species randomly encountering each other, play an important role in shaping ecological networks [49–51]. This neutral component of interactions can be incorporated into predictions of interactions by adding data on species' abundances [52]. Although ecological and life history traits can influence species abundances [38, 53], abundance is usually considered to also result from neutral processes [54–57]. While species traits may determine which interactions are possible (e.g., 'forbidden links'; [58]), the occurrence and frequency of interactions may be driven by species abundances [49], and both trait distributions and abundances may respond to the environment, driving variation in interactions through space and time [53].

Although the above literature suggests that traits, phylogenies and abundances can be useful predictors of species interactions, less is known about the circumstances under which predictions of interactions based on these data are likely to be better (or worse). Emerging research suggests that species characteristics (e.g., trophic generality or biogeographic status; native vs. exotic) may influence the predictive ability of trait- (including phylogenetically-) or abundance- based models [59, 60]. Additionally, research has indicated that predicting network structure at a new habitat can be difficult [51, 53], potentially because species composition and

the interactions that are realised, which in turn can influence species' realised generality or abundance, can vary with habitat type [61]. Furthermore, in the context of biological control, it would be necessary to predict interactions for agents in their introduced location (as an exotic species potentially interacting with a non-target native community) and across a diversity of both natural and crop habitats [53]. Thus, understanding the robustness of predictions to habitat or biogeographic differences and trophic generality would be crucial.

Combining machine-learning and network approaches, if successful, could allow non-target impacts of proposed parasitoid biocontrol agents to be predicted given a list of host species in the recipient community, along with known host-parasitoid interactions and species data (traits and phylogenies) to inform a machine-learning model. Known interactions from either a biocontrol agent's native range or the community to which an agent is to be introduced could be used. Alternatively, if predictions of interactions can be made equally well within vs. across habitat types, network data from a different habitat could be used to inform a predictive model.

In this study, we examine whether network approaches can be used to predict effects (both direct and indirect) of species in parasitoid-host networks, with a view to using these approaches in biological control. However, rather than focusing on biocontrol species only, we focus on developing a proof of concept using an entire host-parasitoid community (only some of which are pests and biocontrol agents), which comprises a large-scale dataset, sampled in two different habitat types (plantation and native forest), from Frost *et al*. [21]. Previous work has shown that indirect effects are important for structuring host-parasitoid interactions in this study system [21], making it suitable for answering our research questions. Specifically, our objectives were to 1) compare two machine-learning techniques (random forest and KNN) for the prediction of (quantitative) host-parasitoid network interactions (i.e. direct effects), using traits and phylogenies to inform the models (which could complement host-range studies currently used to assess direct effects); 2) test whether the ability of these machine-learning techniques to predict species interactions depended on a) species' trophic generality, or b) species' abundances or c) whether the data used to train the predictive model came from the same vs. different habitat type as that for which the predictions were being made; and 3) determine whether these machine-learning-predicted interactions, when combined into networks, can be used to predict indirect effects (specifically apparent competition), which could then be applied to predict non-target effects of biocontrol agents.

## Methods

### Study system

We collected quantitative host-parasitoid interaction data from 16 training sites (8 from native forest and 8 from plantation) and 32 testing, i.e. validation, sites (16 from native forest and 16 from plantation) in the Nelson/Marlborough region of New Zealand (further details of sampling methods are provided in Frost *et al*. [21] and Peralta *et al*. [62]). Each training site consisted of 2 transects; an edge (10 m towards the forest interior from the edge between the two forest types, and running parallel to the edge) and interior (400–500 m from the forest edge). Each test site consisted of 1 edge transect (from its respective forest type). The test sites did not include interior transects; however, we chose to include both edge and interior transects in the training sites to use all available data. At each transect (50 m x 2 m), all foliage-dwelling Lepidoptera larvae ('hosts') were collected from plants and reared in the laboratory until they developed into adults or parasitoids emerged (Hymenoptera, Diptera and Nematoda; though we excluded Nematoda parasitoids from all our analyses as the traits we measured, see 'Traits and phylogenies' below, were not suited to these species). Parasitoids and hosts were then

identified to species level where possible using available taxonomic information [63–66] and expert assistance (J. Dugdale, J. Berry, and R. Schnitzler), and otherwise to morphospecies. For parasitoids, morphospecies were validated as species using molecular methods. The training sites were sampled seven times over two summer seasons (season 1: December 2009, January, February 2010; season 2: October, November 2010, January, February 2011). To determine whether predictions of apparent competition were equally effective for large experimental and small natural changes in host abundances, host abundance was experimentally reduced in half of the plantation test sites by aerial spraying with a caterpillar-specific biopesticide in the middle of sampling. Test sites were sampled twice before this host reduction treatment (time *t*, October, November 2010) and twice after (time *t+1*, January, February 2011). Because Frost *et al.* [21] found that the predictive ability of apparent competition did not differ across the host-spray treatment, we did not include this variable in our analyses.

## Traits, phylogenies, and abundance

We collected traits of host and parasitoid species that have been identified as being useful for predicting interactions: body size, trophic generality (measured as normalised number of food resource species, termed 'normalised degree' in network studies; ND), biogeographic status (native vs. exotic), and phenology. Note that ND was measured for parasitoids based on their interactions with hosts and for hosts based on their interactions with their plant hosts, rather than parasitoids. We chose these traits because body size is correlated with many important characteristics [67] and is often the best predictor of species (including host-parasitoid) interactions [45], while species generalism tends to be conserved across native and alien ranges [68], and recent evidence suggests that niche processes for species interactions (and therefore predictions based on traits and phylogenies) are weaker for generalists than specialists, and for exotic species than native species [59]. Additionally, phenology plays a role in determining interactions [56, 69]; we therefore calculated two aspects of phenology for both host and parasitoid species: abundance of each species in each month and phenological overlap between each host-parasitoid pair (see 'Phenology' in S1 Appendix). Because phylogenies can also help to explain interaction patterns [44, 46] by capturing variation in difficult to measure traits, we constructed host and parasitoid phylogenies to include along with the traits in the prediction of interactions (see 'Traits and phylogenies' in S1 Appendix). Additionally, abundance (a neutral process) can be important for determining interactions [55, 56]; we therefore summed abundance across sites and sampling dates for both host and parasitoid species, to be used in a later part of the analysis (see 'Abundance' in S1 Appendix).

## Predicting direct effects

We compared two machine-learning techniques (random forest and KNN) for the prediction of quantitative host-parasitoid interactions (i.e. direct effects). Training site data were used to inform the models and test-site data (at time *t*) were used to test (i.e. validate) the models (Fig 1).

   To predict interactions at new locations (or for new species such as biocontrol agents), the random-forest model learns (from the training data) which host and/or parasitoid trait (or phylogenetic) combinations are likely to correspond to an interaction occurrence. The algorithm then uses these rules, along with the traits (or phylogenetic position) of new species, to predict the probability of interaction occurrence between new combinations of host and parasitoid species pairs. These interaction probabilities could be used in risk assessment if it can be demonstrated that they correspond with higher observed interaction frequencies (which we test below). Random forest therefore assumes that the strength of trait (or phylogenetic)

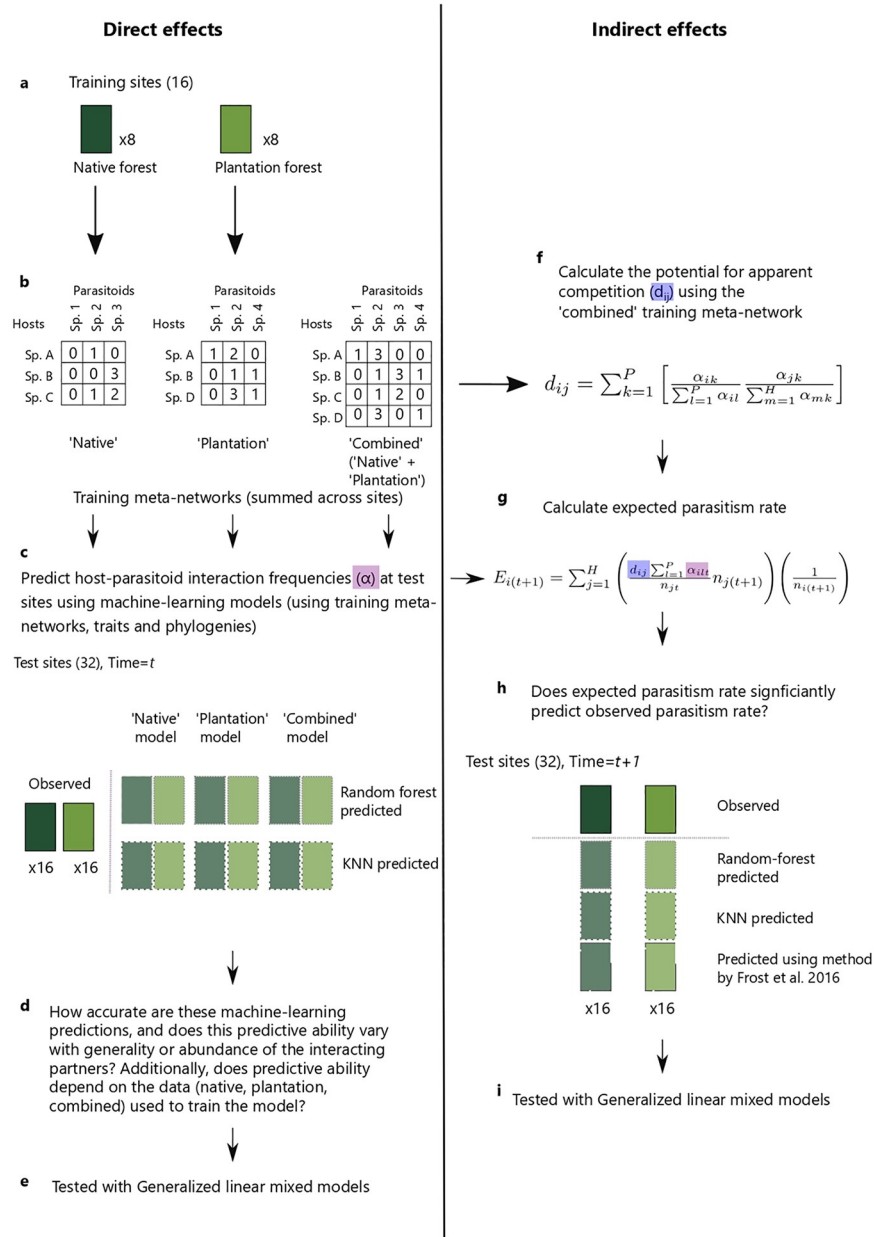

**Fig 1. Summary of methods.** (a) Lepidopteran larvae ('hosts') were collected from eight native forest sites and eight plantation forest sites (we call these 'training sites' as data from these sites were used to train the models). (b) Hosts were identified and reared to determine parasitism rates and identities. We created three quantitative training meta-networks ('native', 'plantation', 'combined') by pooling interaction data across native sites, plantation sites, and all sites, respectively. (c) We used machine-learning models (random forest and KNN), informed by training meta-networks, traits and phylogenies, to predict interaction frequencies at test sites (16 native forest and 16 plantation forest) at time $t$. We created a separate model for each of the three training meta-networks, for both machine-learning approaches (i.e., 6 models: random forest/KNN 'native'/'plantation'/'combined'). (d, e) We then tested whether the ability of these machine-learning models to predict interaction frequencies depended on species generality or abundance, and whether predictive ability depended on the data (i.e. native, plantation, combined) used to inform the model. (f) We then calculated the potential for apparent competition, $d_{ij}$, between each pair of host species in the training data (using both native and plantation sites data). (g) We calculated expected parasitism rates (E) for host species at time = $t+1$ test sites using these $d_{ij}$ values, along with predicted attack rates ($\alpha$) calculated from our machine-learning predicted networks at time = $t$ test sites, and abundance ($n$) from the test sites. We calculated expected parasitism rates (E) for both random forest and KNN. Additionally, we calculated expected parasitism rates using a data-based approach used by Frost *et al.* [21]. We tested whether expected parasitism rates could significantly predict observed parasitism rates at time = $t+1$ test sites (h, i).

matching between species determines, in part, the likelihood of an interaction between them. In contrast to random forest, which incorporates matching of parasitoid traits (or phylogenetic position) to those of hosts, KNN uses the host-use preferences of parasitoid species (from the training data) with similar traits (or phylogenetic position) to recommend (i.e., predict) interaction partners and interaction frequencies for new parasitoid species. This is analogous to Netflix recommending films to users based on the preferences of similar users [47]. Thus, in KNN, parasitoid traits are used to determine similarity among parasitoid species, rather than to match with host traits as in random forest (see 'Machine-learning techniques' in S1 Appendix).

**Model variants with different training data.** When making predictions about the impacts of potential biocontrol agents, it is possible that interaction and/or trait data will not exist for the specific recipient habitat. Therefore, to determine whether the ability of a machine-learning technique (random forest or KNN) to predict interactions differed when training data came from the same vs. different habitat than the test habitat, we built three models ('native', 'plantation', 'combined') for each technique, using data (i.e. pooled 'meta-networks') from the training sites of each habitat category. We created one host-parasitoid training meta-network for each of the three models, by pooling data across training sites (of a single forest type, native or plantation, for the 'native' and 'plantation' models, respectively, and all training sites, i.e., both forest types, for the 'combined' model) and sampling dates (see 'Training meta-networks' in S1 Appendix). Because the random forest was predicting the probability of an interaction occurring, whereas KNN predicts a weight of the interaction (i.e. frequency), we trained the random-forest models on the binary (occurrence vs. not of each interaction) meta-networks and trained the KNN models on the weighted meta-networks (though for both approaches a quantitative interaction network was predicted).

For KNN, recommendations (i.e., predictions) are based on host use by parasitoid species in the training data, so only host species present in the training data will be recommended to parasitoid species in the test data. In contrast, host species only found in the test data, hereafter called 'new' host species, will never be recommended to a parasitoid—these host species are analogous to new films that no one has seen or rated yet. To overcome this problem, we included 'new' host species into the training meta-networks (only for KNN models because random forest does not suffer from this problem) by recommending parasitoid species from the training data to each 'new' host species, using KNN. In this case, we recommended parasitoid species to hosts by using host traits and phylogenetic position to determine their nearest neighbours (i.e. the host species most similar to them) (see 'KNN–adding 'new' host species' in S1 Appendix).

We tested the predictions of each model on all test sites where parasitoids emerged from any collected host at the first-time step, $t$ (i.e., before the host spray treatment: 14 sites from native forest and 15 sites from plantation). Test sites at which no parasitoids emerged at time $t$ were excluded from all analyses (N = 3). For each test site, we predicted an interaction frequency (for KNN; see 'KNN–predicting interactions at test sites' in S1 Appendix) or a probability of interaction occurrence (for random forest) between each host-parasitoid species pair present at the site. If a parasitoid species was found in both the training and testing data, it was not used to inform predictions about its own interaction partners in the KNN algorithm (i.e. it was not used as a neighbour of itself); this decision was made to be conservative, because interaction data in the new location would likely not be available for a proposed control agent.

We performed a grid search to find the optimal hyperparameters (including which traits to include) for the random-forest and KNN models (see 'Tuning hyperparameters' in S1 Appendix). Random forest automatically weights traits by their importance (i.e. less informative traits contribute less to predictions). To determine which traits were most important for predicting

host-parasitoid interactions using random-forest models, we used the 'feature importances' function (which uses Gini importance) from Python's Scikit-learn v 0.23.1 package [70]. In contrast to random forest, in KNN all traits contribute equally to measures of species' similarity. For this reason, unimportant traits must first be removed, as retaining them in the model can decrease predictive ability [71]. In our case of predicting host-parasitoid interactions using KNN models, we found that body size, phenology, and ND (normalised degree; a measure of generality) were useful parasitoid traits, whereas phylogeny was removed during our parameter search process (see 'Best hyperparameters for KNN' in S1 Appendix). Additionally, we found that all host traits (i.e., body size, phenology, phylogeny, host ND and biogeographic status) were useful (and thus retained in the models). For the random-forest models, parasitoid traits in decreasing order of importance were ND, phylogeny, body size, phenology, while host traits in decreasing order of importance were phenology, phylogeny, body size, ND, and biogeographic status (for both hosts and parasitoids this is the average importance across all three models: 'native', 'plantation', 'combined') (Table 20 in S1 Appendix). All random-forest and KNN analyses were performed in Python (version 3.8.3). We used the 'RandomForestClassifier' function from the Scikit-learn package [70] to perform the random-forest analyses, and used base Python for KNN.

## Predicting indirect effects

To predict indirect interactions (apparent competition), we used the quantitative interaction network (i.e. direct interactions at each test site at time $t$) predicted by each model (random-forest or KNN) trained on the 'combined' data (native forest and plantation data). This quantitative interaction network can be used to project a network of indirect linkages among hosts that share parasitoids. We can then predict how attack rates on each host should change as a function of changes in the abundance of host species that share parasitoids with that host (i.e. the indirect effect of 'apparent competition'). Therefore, we predicted how patterns in attack would change if the indirect (parasitoid-sharing) links among hosts matched those predicted by the machine-learning methods. We then went on to validate how these predicted changes in attack rate matched real changes observed (between times $t$ and $t+1$). Note that the machine-learning predicted networks are based only on species traits and phylogenies (i.e. information that captures niche processes), and abundance is included later into the predictions of indirect effects (see Eq 1 below). We used these 'combined' models because, in a biocontrol context, available data may not be specific to a target habitat, but rather aggregated from different sites and habitats.

Predicting indirect effects involved three steps. First, we used the 'combined' training meta-network (see above) to calculate a regional quantitative measure of shared parasitoids, also known as the potential for apparent competition, $d_{ij}$, using Equation 6 from Müller *et al.* [23]; see 'Measure of shared parasitism' in S1 Appendix. This $d_{ij}$ value measures, for every pair of host species in a community, the proportion of parasitoids attacking host species $i$ that recruited from host species $j$ [23]. We used the PAC function from the bipartite v 2.15 R package [72] to calculate $d_{ij}$. Second, for each machine-learning approach (random forest and KNN), we used this measure of shared parasitism ($d_{ij}$), along with random-forest- (or KNN)-predicted initial attack rates ($\alpha$ in Eq 1 in S1 Appendix, see also 'Calculating $\alpha$ from predicted networks' in S1 Appendix) by each parasitoid on each host from time $t$ at the test sites combined with known changes in host abundances (the change in abundance of each host species from sample $t$ to $t+1$, in the site for which predictions are being made) to calculate the expected parasitism rate (E) at time $t+1$ of host species $i$ for each test site. Parasitism rate is the number of parasitism events divided by the number of hosts sampled. We used Equation 3 from Frost

*et al.* [21] to calculate E:

$$E_{i(t+1)} = \sum_{j=1}^{H} \left( \frac{d_{ij} \sum_{l=1}^{P} \alpha_{ilt}}{n_{jt}} n_{j(t+1)} \right) \left( \frac{1}{n_{i(t+1)}} \right) \tag{1}$$

where $n$ is host abundance, $t$ is the first-time step of data (before experimental host reduction in half of the sites, with $t+1$ being after), $\alpha$ is the link strength (that is, number of attacks), $i$ and $j$ are a focal host species pair, $H$ is the total number of host species, and $l$ is all parasitoid species, from 1 to $P$ (the total number of parasitoid species).

Finally, we tested whether this expected parasitism rate significantly predicted the observed parasitism rate of each host species $i$ at time $t+1$ test sites (see equation 2 in S1 Appendix for how observed parasitism was calculated).

The logic of this approach is that a bipartite network of interactions, where links represent attack rates between hosts and parasitoids (the nodes), can be projected into a unipartite network with only hosts as nodes and links representing the proportional sharing of parasitoids ($d_{ij}$). Changes in abundance of hosts in this unipartite network should induce proportionate (and thus predictable) changes in the abundance of their parasitoids and cause a proportionate change in attack rates on any other hosts that share those parasitoid species (i.e. an indirect effect). Frost *et al.* [21] showed that these expected rates (E) significantly predicted observed parasitism rates, even when within-habitat intraspecific contributions (that is, delayed density dependent parasitism) were excluded from these expected rates, indicating that the indirect effects present could not simply be explained by within-habitat delayed density dependence. This method for predicting indirect effects has thus been experimentally validated using this dataset. Our aim here was to test whether the machine-learning-predicted networks (trained on data from different sites) could be used instead of observed networks to predict these indirect effects.

The number of host species for which we were able to calculate an expected parasitism rate at each site was much lower than the number of species collected at each site. This difference occurred because we could only calculate an expected parasitism rate for host species within a site that were collected at both time steps and had a non-zero predicted attack rate ($\alpha$) at the first-time step ($t$, see Eq 1 above). However, for both machine-learning approaches, we used the potential for apparent competition (via the sharing of parasitoids) with every other host species that was collected in the site, to calculate the expected parasitism rates for these species. Therefore, predictions of expected parasitism rates for these species were based on data from the whole network. The numbers of species within sites for which it was possible to calculate an expected parasitism rate were 91 and 84 for the random-forest and KNN approaches, respectively.

## Statistical analyses

**Can random forest and KNN predict interaction frequencies and does this predictive ability vary with species' generality and abundance, and whether the model was trained on data from the same vs. different habitat?** To test the possibility that random-forest-predicted probability and KNN-predicted-frequency can predict observed interaction frequencies, and to compare the predictive ability of each random forest with that of its corresponding KNN model ('native', 'plantation', or 'combined'), we fitted two Poisson generalised linear mixed effect models using observed interaction frequency (from the test sites at the first time step) as the response variable. We included random-forest-predicted probability as a fixed effect in the first model and KNN-predicted interaction frequency in the second model. Including just these predictors in the models allowed us to compare the two machine-learning methods directly, and we then added further predictor variables as a separate step to test our other hypotheses (see below). This process was repeated two additional times for a total of six models ('native', 'plantation' and

'combined' for both random-forest and KNN predicted probability and frequency, respectively). For each model, we scaled and centred the fixed effect (by subtracting the mean and dividing by the standard deviation such that the scaled variable would have a mean of zero and a standard deviation of one) to help with model convergence. In all full models, we included parasitoid identity as a random factor (such that predicted and observed frequencies for a given parasitoid would be 'paired'), and a random slope for random-forest-predicted probability (or KNN-predicted-frequency), to allow this slope to vary across parasitoid species. We also included site as a random factor, to account for the structural non-independence of interactions across sites, though site explained little variance because it covaried strongly with parasitoid composition. We used parasitoid identity because estimating the random slope required multiple interactions per parasitoid, whereas using host-parasitoid pair (see 'Model set 1' in S1 Appendix) would have resulted in a model in which the number of observations was less than the number of random effects. The R formulas for these, and all subsequent, models are included in S1 Appendix (Tables 21–24 in S1 Appendix).

Fitting the model six times (three times for each machine-learning method) increases the probability of Type I error. Although sequential Bonferroni correction is often used to account for performing multiple statistical tests, it has been argued to be too conservative [73, 74]. We therefore used a Bernoulli process to calculate the probability of each fixed effect being significant by chance alone, given the number of tests (six) performed [73]. We use this Bernoulli process (for fixed effects) for all subsequent analyses that involve multiple statistical tests.

The prediction of interaction frequencies (which is needed, in turn, to predict indirect effects) represents a stricter test than the prediction of interaction occurrence, as it requires not only the correct prediction of interactions, but also of their strength. However, if this information is not needed, the machine-learning-predicted probabilities and frequencies can be converted into binary values to predict interaction occurrence, so we also explored such predictions of interaction presence/absence (see 'Tests on the prediction of interaction occurrence' in S1 Appendix).

For all analyses, we selected the best-fitting model by first selecting the optimal random-effect structure [75]. We ran the full model (described above), as well as models with all combinations of random effects, and selected the model with the lowest AIC. Then, using this optimal random-effect structure for each model, we selected the best-fitting fixed effects structure by using the dredge function (from the MuMIn v 1.43.17 package [76]), which calculates AIC for the full model as well as all possible simpler models.

To compare each random forest with that of its corresponding KNN model, we calculated AIC for each best-fitting model. We primarily use this model selection to identify variables that significantly contribute explanatory power, and caution against the mixing of model selection and frequentist statistics [77]. However, for readers accustomed to P values, these are presented in the results.

To determine whether the ability to predict interaction frequency (using machine-learning models) varied with species generality (measured as normalised degree, ND) or according to whether the model was trained on data from the same vs. different habitat, and also to test whether predictions could be improved by including information on species' abundances, we fitted the same set of models as described above, but with additional predictor variables. Specifically, in addition to random-forest-predicted probability or KNN-predicted interaction frequency, we also included parasitoid ND and host ND as interacting fixed effects (i.e. this model included all possible interactions among these three fixed effects), along with host abundance, parasitoid abundance, and forest type (of the test site) as fixed-effects interacting with predicted probability or predicted frequency (i.e. this model also included all possible interactions among these four fixed effects). Because we did not have a specific hypothesis for the forest type x ND interactions (or the abundance x ND interactions), we did not include these

interactions in the models. We followed the same model selection process as described above. Abundance and generality can be related; therefore it could be possible for model selection (described above) to return a model with only one of these variables if they share a high amount of variance. To determine the amount of collinearity between these two variables (for both hosts and parasitoids), we calculated the proportion of models (with predictors that were any possible subset of those in the full model) with delta AIC<2 of the best model which contained each of the four variables (host or parasitoid ND or abundance). We found that for all 6 models within a delta AIC of 2, all ND and generality variables that were retained in the best-fitting model were also retained in all candidate models with delta AIC<2 of the best-fitting model. The one exception to this was parasitoid abundance which was retained in the best-fitting KNN 'native' model, and was included in 80% of candidate models with delta AIC<2. Conversely, in all cases where a ND or generality variable was not included in the best-fitting model, this variable was retained in at most 25% of the models with delta AIC<2 (and in one case retained in 50% of the models with delta AIC<2). This indicates that ND and generality are not highly collinear in our study system.

**Can we predict indirect effects using machine-learning (random forest and KNN) informed methods?** To determine whether expected parasitism rate can predict abundance-driven changes in the observed parasitism rate (at time *t+1* test sites), and therefore whether indirect effects can be predicted, we fitted a binomial mixed effect model including observed parasitism rate (a proportion) as the response variable. We included expected parasitism rate as a fixed effect and site as a random effect (to control for the non-independence of interactions at a given site). We did not include forest type (native forest vs. plantation) of the test sites as a fixed effect in this model, because Frost *et al.* [21] found that the ability to predict observed parasitism rates did not differ across this variable in this study system. We scaled and centred the expected parasitism rate fixed effect to help with model convergence.

We fitted a separate version of this model to the predictions of expected parasitism rate (see Eq 1 above) from each machine-learning approach. For each version of the model, we followed the same model selection process as described above.

**Do predictions of indirect effects differ depending on whether observed vs. predicted network data are used to calculate the potential for apparent competition ($d_{ij}$)?** As an additional supplementary analysis, we tested whether predictions of observed parasitism rates differed depending on the network used to calculate the potential for apparent competition ($d_{ij}$). The rationale for this analysis was that the observed training data are only a sample of the true network, and thus may miss interactions that are important for driving indirect effects. Thus, it is possible that predicted networks better capture these unsampled interactions. Conversely, it is also possible that errors in the predicted networks could lead to poorer estimation of $d_{ij}$ and resulting predictions of indirect effects. For this analysis, we re-ran the above analyses using $d_{ij}$ values calculated from the machine-learning predicted networks, rather than the observed training networks. Overall, this approach did not markedly improve the prediction of indirect effects (marginal $R^2$ increased by 0.008), but for completeness we present it in the S1 Appendix (methods and results in S1 Appendix: see 'Tests using $d_{ij}$ calculated from predicted network data' in S1 Appendix).

**Do predictions of indirect effects based on machine-learning-predicted networks differ from those based on observed network data?** To compare the random-forest and KNN approaches with each other, and with a purely data-driven approach used by Frost *et al.* [21], we re-ran the above analyses (but not the additional analyses using machine-learning predicted $d_{ij}$ values) using only the subset of host species for which it was possible to calculate an expected parasitism rate with all three methods (random forest, KNN, and data-based as used by Frost *et al.* [21], N = 20). To calculate expected parasitism rate using the data-based

approach used by Frost *et al.* [21], we calculated the initial attack rates ($\alpha$) from the observed time *t* networks at the test sites and calculated $d_{ij}$ from the 'combined' training meta-network. This is the same approach used by Frost *et al.* [21], except that we did not use habitat specific $d_{ij}$ values and used both edge and interior sites from the training data (rather than just edge sites). We made these changes to the data used because, in a biological control context, site-specific data are less likely to be available, and also to increase the amount of data used to inform predictions in the machine-learning approaches. Additionally, we did not include the seven host species for which we could not determine their biogeographic status (see 'Biogeographic status' in S1 Appendix). As before, we performed model selection for each of the three approaches (random forest, KNN and data-based) to find the best-fitting model following the same process described above, except that we used AICc (which is more suited to small sample sizes than AIC), to select the best-fitting model. To compare the three approaches, we calculated the AICc of the best-fitting model for each approach. We visually inspected these models for outliers and, where present, we calculated the leverage value of all datapoints and removed those with high influence (i.e. outliers [78]). Outliers were present in these models predicting indirect effects, though in all cases the results with the outliers removed were qualitatively the same as the results with the outliers included. Note that we did not remove outliers from the models comparing the three methods (random forest, KNN and data based as used by Frost *et al.* [21]) for predicting indirect effects on the common dataset (N = 20) to ensure that the number of datapoints was kept constant among models to allow model comparison, though in all cases excluding outliers did not qualitatively change the results.

All statistical analyses were performed using R v 4.0.1 [79]. We used the lme4 v 1.1.23 package [80] to fit mixed effects models. We used the r.squaredGLMM function (with the delta method) from the MuMIn v 1.43.17 package [76] to calculate marginal and conditional values of $R^2$ for the mixed effects models and the rsquared function from the piecewiseSEM v 2.1.0 package [81] to calculate $R^2$ for the generalised linear models. For all models we checked for overdispersion, and where present we included an observation-level random factor [82], and in all cases this resulted in a model that was not over-dispersed (i.e., the ratio of the sum of squared Pearson residuals to residual degrees of freedom was <2 and non-significant when tested with a Chi-squared test [83]).

## Results

### Can random forest and KNN predict interaction frequencies and does this predictive ability vary with species' generality and abundance, and whether the model was trained on data from the same vs. different habitat?

We tested whether machine-learning techniques (random forest and KNN) can predict host-parasitoid interaction frequencies. For the less complex set of models (including only random-forest-predicted probability or KNN-predicted frequency), we found that random forest was more successful than KNN at predicting interaction frequencies. Random-forest-predicted probability was significantly positively related to observed interaction frequency in all three models ('native', 'plantation', 'combined'), and captured a reasonably high amount of variance (marginal $R^2 > 0.338$ for all three models) (Fig 2A; Tables 1 and 3 in S1 Appendix). In contrast, KNN-predicted interaction frequency was not retained in any of the three KNN models (Fig 2B; Tables 2 and 3 in S1 Appendix). There was a low probability of the predicted-probability fixed effect being significant in 3/6 models by chance (p<0.001, calculated using the Bernoulli process), indicating that the results are unlikely to be due entirely to type I error. For all three models ('native', 'plantation', 'combined'), the random-forest model predicted observed interaction frequency better (i.e. lower AIC) than the respective KNN model (Table 3 in

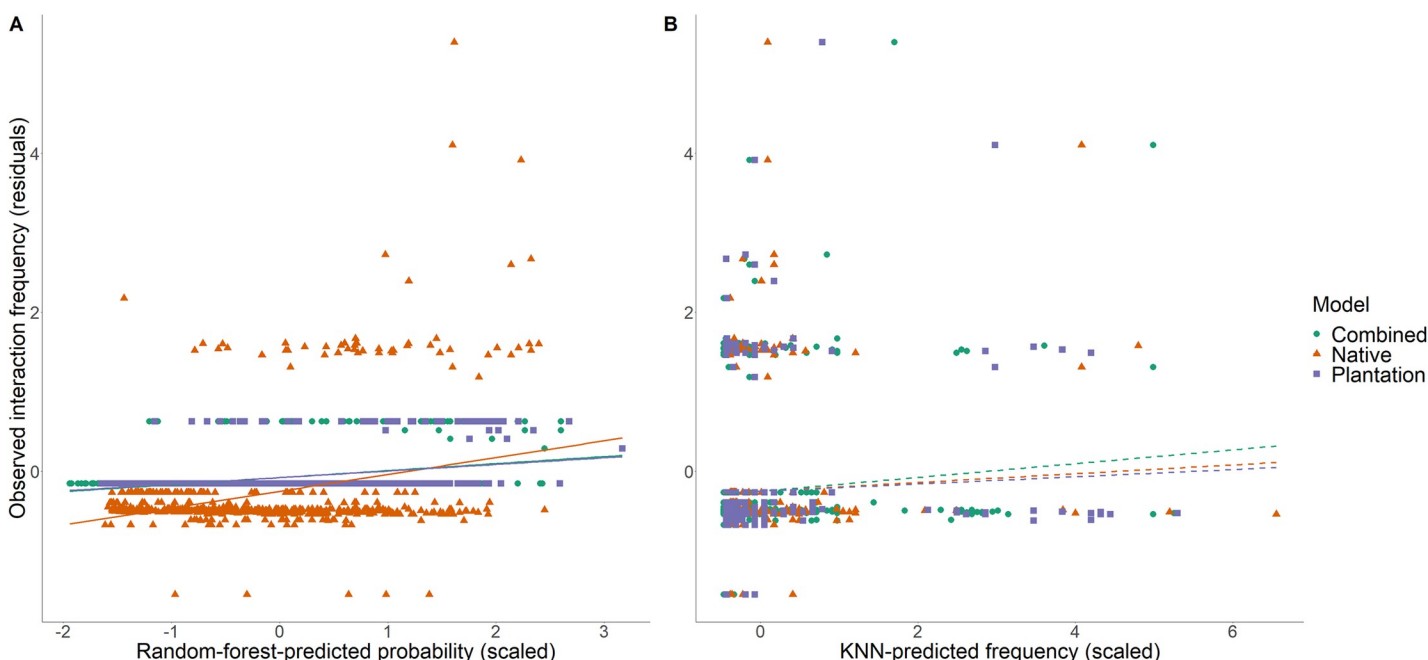

**Fig 2. Predictions of observed interaction frequency.** (A) Random-forest-predicted probability significantly predicted observed interaction frequency for all three random-forest models ('combined', 'native', 'plantation'), whereas (B) KNN-predicted frequency was not retained in any of the best-fitting models (tested using Poisson linear mixed models, and dashed lines represent non-significant relationships). Deviance residuals of the best-fitting model excluding the predictor on the x-axis are plotted, to better reflect the partial effect of the predictor. Each point represents a host-parasitoid species pair present within a site at the first-time step (*t*) test sites.

S1 Appendix). Additionally, for 2/3 models ('native' and 'combined'), the random-forest model captured a greater amount of variation in observed frequency (higher conditional $R^2$ values) than the respective KNN models. All best-fitting models included parasitoid identity as a random factor, and a random slope for random-forest-predicted probability (or KNN-predicted frequency) (except for the random-forest 'plantation' and 'combined' models which only included an observation-level random effect, along with parasitoid identity as a random factor in the 'combined' model).

For the second set of models, testing whether the ability to predict interaction frequency differed for generalists vs. specialists, whether predictive ability depended on species' abundances, and whether predictions can be made equally within vs. across habitats, we found that the predictive ability of KNN, but not random forest, depended on the generality (ND) of the interacting partners (Fig 3; Tables 4 and 5 in S1 Appendix). We also found that predictive ability depended on species' abundances, but not on the source of the training data, for both random forest and KNN.

There was a significant positive relationship between random-forest-predicted probability and observed interaction frequency in all three models (i.e. higher predicted probabilities corresponded to higher observed frequencies), qualitatively the same as for the less complex models (see above). In contrast, predicted frequency was only significantly related to observed interaction frequency in the 'native' KNN model (for the less complex models, predicted frequency was not retained in any of the three KNN models).

The host ND fixed effect was included in all three KNN models, along with parasitoid ND in the 'native' and 'combined' KNN models; however, neither of these variables were included in any of the best-fitting random-forest models. There was a significant positive three-way interaction between predicted frequency, host ND, and parasitoid ND in the KNN 'native'

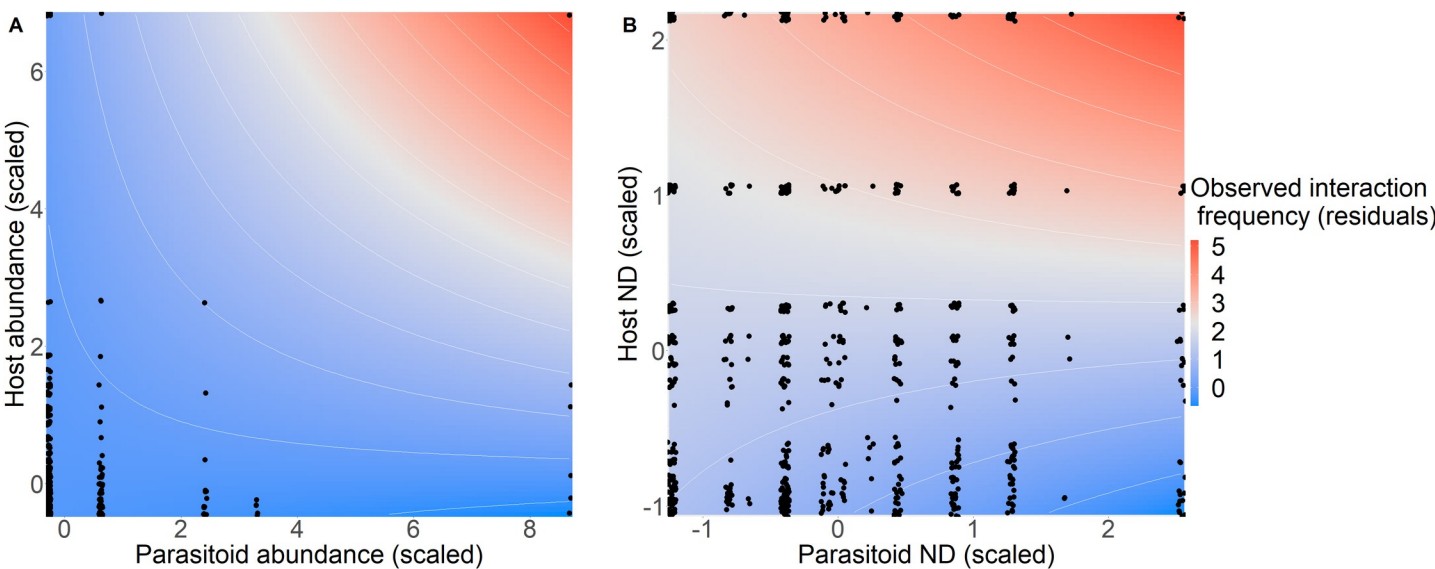

**Fig 3. Overall, machine-learning predictions of observed interaction frequency varied with the generality (measured as normalised degree; ND) of the interacting partners for KNN, but not random-forest models, and depended on the abundance of the interacting partners for both models.**

model, such that when ND of both host and parasitoid were high, the slope of the relationship between predicted frequency and observed interaction frequency was more positive than expected based on additivity of the main effects alone. However, the presence of negative (and significant) two-way interactions between predicted frequency and ND (of both host and parasitoid), along with a negative host ND main effect, which outweighed the positive parasitoid ND main effect, meant that interactions involving a single generalist partner would tend to occur at lower frequency and have a weaker relationship between predicted and observed frequency, but these effects would be weakened when both partners were generalists. Although the predicted frequency x host ND x parasitoid ND interaction was retained in the KNN 'combined' model, it was not significant in this model. However, there was a significant negative two-way interaction between predicted frequency and host ND in this model, along with a negative host ND main effect, indicating that when host ND is high, the relationship between predicted frequency and observed interaction frequency was more negative than expected. In the KNN 'plantation' model, host ND did not interact with predicted frequency (and parasitoid ND was not retained in this model).

There was a significant negative three-way interaction between predicted probability (or predicted frequency), host abundance, and parasitoid abundance in the 'plantation' and 'combined' models for both random forest and KNN, meaning that the relationship between predicted probability (or predicted frequency) was more negative when the abundance of both interacting partners was high. However, the presence of two-way interactions involving predicted probability/frequency and abundance meant that high abundance of both interacting partners affected predictions differently to when the abundance of only one of the interacting partners was high. Because our data contain few species with very high abundance values, caution should be taken when interpreting these results.

There was a significant three-way interaction between predicted-frequency and generality of the interacting partners (host and parasitoid) in the KNN 'native' model. Additionally, there was a significant three-way interaction between predicted-frequency (or -probability) and abundance of the interacting partners for 4/6 models. Two are presented here to demonstrate the main patterns: (A) KNN 'native' model (which had a significant positive predicted

frequency x host ND x parasitoid ND interaction), (B) random-forest 'combined' model, which had a significant host abundance x parasitoid abundance x predicted probability interaction (as did the 'plantation' random-forest model and the KNN 'plantation' and 'combined' models). However, the limited data between species with extreme abundance values, means this result should be treated with caution. Deviance residuals of the best-fitting model excluding the host and parasitoid ND (or abundance) variables are plotted as colour contours, to indicate whether observed interaction frequencies are higher (red) or lower (blue) than expected based on machine-learning predictions, showing how predictions vary with generality (or abundance) of the interacting partners. Each point represents a host-parasitoid species pair present within a site at the first-time step ($t$) test sites.

For all random-forest and KNN models, the relationship between interaction probability (or predicted frequency) and observed interaction frequency did not differ between the two forest types (i.e. the forest type fixed effect did not interact significantly with predicted-probability or -frequency in any of the 6 models), implying that predictions of interaction frequency (based on random-forest-predicted probabilities or KNN-predicted frequencies) are equally effective in both forest types and do not depend on the source of training data.

No random effects were retained in any of the best-fitting models for both random forest and KNN (except for the KNN 'native' model, which included the parasitoid identity random effect, which explained some variance; $R_c^2 = 0.147$). The fixed effects explained more variation in interaction frequency in all three random-forest models ($R^2 > 0.23$) than their respective KNN models ($R^2 > 0.11$).

The above results contrast with the analyses for predicting interaction occurrence (see 'Tests on the prediction of interaction occurrence' in S1 Appendix). For both random forest and KNN, all three models ('native', 'plantation', 'combined') predicted interaction frequencies better than binary interaction occurrences (models predicting frequencies had higher marginal and conditional $R^2$ values than their respective, i.e. 'native', 'plantation', 'combined', model predicting interaction occurrences). This suggests that raw predicted probabilities (or frequencies) are more useful for predicting interactions than assigning presence vs. absence of an interaction occurrence based on an arbitrary probability/frequency threshold.

## Can we predict indirect effects using machine-learning (random forest and KNN) informed methods?

For both random-forest and KNN approaches, there was a significant positive relationship between expected parasitism rate and observed parasitism rate (random forest: z value = 2.72, p = 0.006, df = 85; KNN: z value = 2.80, p = 0.005, df = 77) (Fig 4; Table 7 in S1 Appendix). The probability of this relationship being significant in both models by chance was very low (p<0.001, calculated using the Bernoulli process). Although these models did not explain a large proportion of the variance (random forest marginal $R^2 = 0.155$, KNN-marginal $R^2 = 0.0919$), in nature we would expect many processes beyond just trait matching to determine field parasitism rates, though we take the significant effects of both RF- and KNN-predicted parasitism rates as evidence that these variables were informative (Table 8 in S1 Appendix). The site random effect was retained in both models and also explained a reasonable amount of variance (random forest: conditional $R^2 = 0.403$, KNN: conditional $R^2 = 0.325$).

## Do predictions of indirect effects based on machine-learning-predicted networks differ from those based on observed network data?

The three methods (random forest, KNN and data-based, as used by Frost *et al.* [21]) did not differ in their ability to predict observed parasitism rates (difference in AICc<1) (Table 9 in S1

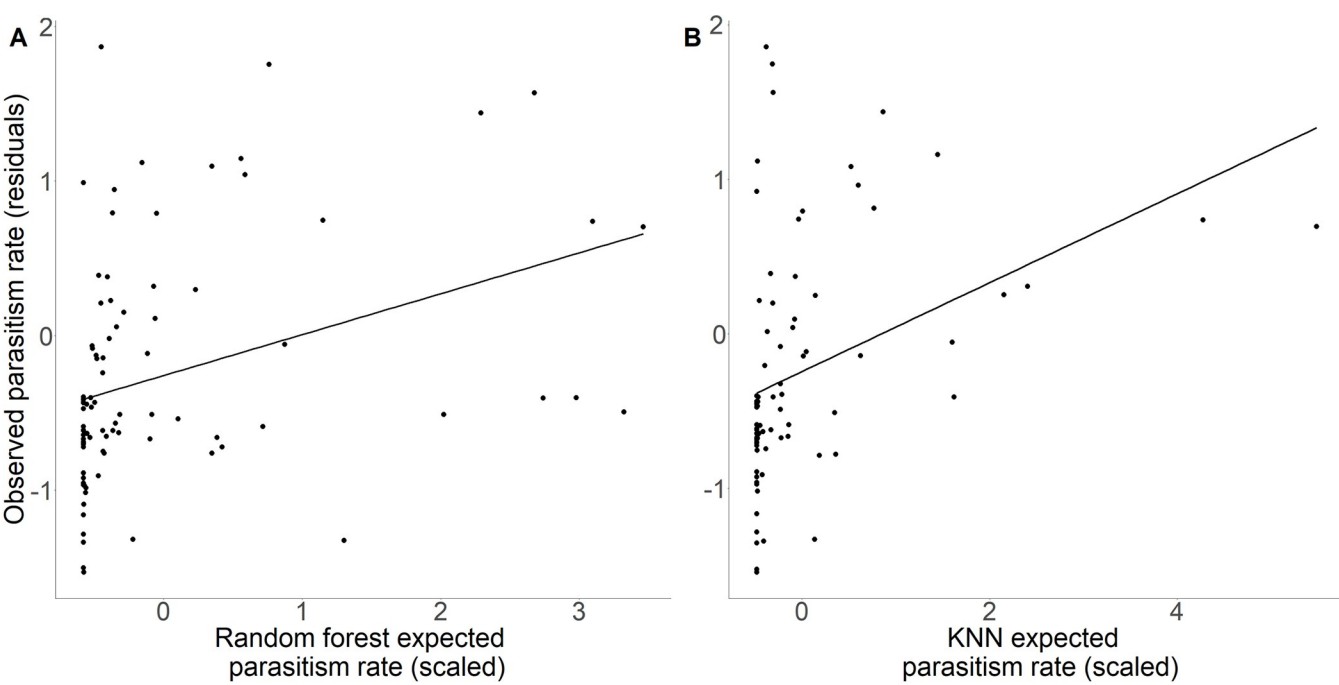

**Fig 4. Predictions of observed parasitism rate.** (A) Random-forest- and (B) KNN- expected parasitism rate both significantly predicted observed parasitism rate. Residuals of the best-fitting model excluding the predictor on the x axis are plotted. Each point represents a host species within a site in the second-time step (*t+1*) test sites. Three and four outliers, respectively, were removed from (A) and (B), though the results with the outliers included were qualitatively the same in both cases.

Appendix). In fact, with the reduced dataset (N = 20; data that could be used for all three analyses), expected parasitism rates did not even significantly predict observed parasitism rates for the random-forest and KNN approaches, contrasting with the results with the full dataset (see above), likely due to reduced statistical power (Table 10 in S1 Appendix). Similarly, expected parasitism rate was not retained in the best-fitting model for the data-based approach. The site random effect was retained in all best-fitting models and captured a reasonable amount of variance (conditional $R^2$ = 0.344 for all models).

## Discussion

The increasing advocacy of biocontrol agents to suppress pest species, combined with their potential to have non-target impacts [10, 11, 84], necessitates the study of methods for assessing the risk of new organisms prior to their release as biocontrol agents [12, 13]. Here, we examined whether network approaches can be used to predict direct and indirect effects of species (e.g., proposed parasitoid biocontrol agents) by comparing the ability of two machine-learning techniques (random forest and KNN) to predict any interaction in the host-parasitoid community. We found that random forest was more successful than KNN for predicting quantitative host-parasitoid pairwise interactions (which could be used to predict attack of non-target host species), using information on a limited set of easily-obtainable traits. In addition, we found that this predictive ability varied with the generality (ND) of the interacting partners for KNN (but not random-forest) models, and depended on species' abundances for both random forest and KNN. Additionally we found that this predictive ability did not depend on the source of training data (i.e., native forest, plantation, or both habitat types combined) for both machine-learning techniques. Further, we found that machine-learning informed methods could significantly predict indirect effects, though with relatively low explanatory power.

## Random forest predicts host-parasitoid interactions better than KNN

The better predictive ability of random forest, compared with KNN (Fig 2), may be explained by the different 'philosophies' of the two methods. While random forest incorporates trait and phylogenetic matching, which are important for determining, and therefore predicting, interactions [37–39, 44, 46], KNN assumes that similar parasitoid species interact with similar host species. However, parasitoid species with similar traits may be dissimilar in their host use, for at least two reasons. First, adaptation to different host plants can lead to insect speciation [85–88], and this may 'cascade' to higher trophic levels, leading to host-associated differentiation in parasitoids [89–92]. Similarly, host-shifts (including the formation of 'host races') can contribute to parasitoid speciation [93, 94], and this parasitoid speciation, by either adaptation or host shifts, may result in closely related parasitoid species that attack different sets of host species. In this way, parasitoid species with the most similar trait values (or that are most closely related) may be dissimilar in their host use, resulting in poor predictions of host use by KNN models. Second, it is possible that parasitoid species deemed as similar, based on the selection of traits we considered (see 'Traits and phylogenies' in Methods), differed in an important trait that we did not include (for example, counter-defences against immune responses of hosts [95]). However, this may be unlikely because we included parasitoid phylogenies, in addition to traits, which capture variation in a variety of measured and unmeasured traits. Additionally, random forest is a more complex algorithm than KNN. Because random forest is based on many trees (each with many branches), random forest is able to account for interactions between variables and non-linearities. This may also have contributed to the better predictive ability of random forest compared to KNN.

## Overall, species' generality and abundance impacts predictions of direct effects

The relative strength of the niche and neutral processes that generate plant-frugivore interactions has been shown to change with species generalism [59], and we similarly found for host-parasitoid interactions that the ability of KNN methods to predict observed interaction frequency depended on the generality (ND) of the interacting partners (Fig 3). Surprisingly, random-forest predictions did not depend on species generality; our random-forest models incorporated trait and phylogenetic matching (two indicators of niche processes), and a dependence of predictive ability on species generality would be congruent with the idea that the strength of niche processes varies with generality [59]. However, we found that predictions of direct effects, by both machine-learning methods, depended on species' abundances, suggesting that strength of niche vs neutral processes may vary with species' abundance.

## Predicting networks within vs. across habitat types

Overall, predictions of host-parasitoid interaction frequencies (and occurrences; see 'Tests on the prediction of interaction occurrence' in S1 Appendix) using machine-learning techniques were equally effective in both habitat types (plantation and native forest). Furthermore, predictions of interaction frequencies (and occurrences) did not depend on the source of the training data, which contrasts with recent research showing that predicting networks across habitats (i.e. using data from one habitat to predict interactions at a different habitat) is difficult [51]. This difficulty is, in part, because interaction frequencies can differ markedly among habitat types (e.g., [96, 97]), arising from both changes in random encounters (due to changes in abundance) and interaction preferences based on niche processes [51]. Congruent with this, Staniczenko et al. [51] found that models that captured systematic changes in interaction

preferences between different habitat types had the best predictive ability, while models based only on interaction data performed poorly. Interaction preferences are likely to differ more with increasing habitat dissimilarity (e.g., [96]). Network data from a more similar habitat (such as the two forest types we compared here) will therefore likely predict interactions better than the forest versus agricultural habitats compared by Staniczenko *et al.* [51], as differences in interaction preferences will likely be smaller. Additionally, similar habitats are more likely to contain species with similar values for traits, which would likely increase the predictive ability of trait-based models. Therefore, the similarity in physical structure (canopy height, etc) and high host and parasitoid species overlap between plantation and native forest habitat may explain why we found no difference in predictive ability of interaction frequencies between the two habitat types. Nevertheless, at face value, the ability to predict equally across habitats and with different training data are positive attributes for biocontrol risk assessment, and suggest that a common dataset (e.g., national databases of species interactions) could be used for risk assessment of interaction frequencies (a pre-requisite for predicting density-mediated indirect effects) in a range of habitats.

### Predicting indirect effects

We found that, although machine-learning informed models significantly predicted indirect effects (i.e., for both random-forest and KNN approaches, expected parasitism rate was a significant predictor of observed parasitism rate) (Fig 4), the explanatory power of both these approaches was reasonably low (approximately 10% of variance). Surprisingly, the prediction of indirect effects was similar (i.e., similar marginal $R^2$ values) to the prediction of direct effects for KNN (Tables 3 and 8 in S1 Appendix), though the highest overall marginal $R^2$ value was observed for the model of direct effects with host and parasitoid generality (ND) and abundance included (Table 6 in S1 Appendix). Together, these results may suggest that the direct effects that were predicted well irrespective of generality (or abundance) were those that were most important for driving indirect effects. Previous work on pollination networks suggests that interactions with a strong signal of trait matching are more spatiotemporally constant and functionally important [98], so it is feasible that the interactions best predicted by machine-learning methods based on traits are those that are least variable and contribute most to indirect effects. These methods are therefore a long way from being able to make accurate predictions of the full strength of parasitoid-mediated indirect effects. However, they represent a promising starting point, and could potentially be used to rank hosts along a continuum from low to high risk of indirect effects. These machine-learning approaches, which use predicted networks rather than observed networks (at time $t$ test sites) to calculate initial attack rates, $\alpha$, (see 'Predicting indirect effects' in Methods), have the potential to capture interactions that are not recorded in the observed networks due to inadequate sampling.

Interestingly, the machine-learning approaches and the data-based approach used by Frost *et al.* [21] did not differ in their ability to predict indirect effects for the common dataset (N = 20) of host species, for which it was possible to calculate an expected parasitism rate with all three methods; this lack of difference was possibly due to reduced statistical power. Moreover, we found that the data-based approach did not significantly predict observed parasitism rate, contrasting the results of Frost *et al.* [21]. This is likely because we used slightly different data to Frost *et al.* [21]. In particular, Frost *et al.* [21] used only edge training sites and explicitly included host habitat (i.e., host species found in different habitats were treated as separate species) when calculating $d_{ij}$ (a measure of shared parasitism), whereas we used both edge and interior training sites and did not explicitly consider host habitat. Species composition and interactions differ between edge and interior sites in this system [99, 100]; thus the inclusion of interior sites may have caused the

different results. Additionally, we removed seven host species (for which we could not determine their host biogeographic status) from all our analyses, and we did not include nematodes because their traits were not comparable to those of insect parasitoids. Collectively, these differences likely explain the incongruence with the findings of Frost *et al*. [21].

Although we found that machine-learning-informed models could significantly predict indirect effects (discussed above), part of the field validation data arose from natural changes in abundance, rather than manipulative additions or removals of individuals. It is therefore possible that an unmeasured factor is correlated with both parasitoid sharing and changes in host abundance or attack rates. For example, habitat preferences of parasitoid species may co-vary with parasitoid sharing, such that the sharing of parasitoids (i.e., apparent competition) is not the cause of the model's ability to predict observed parasitism rates, but rather a correlate. Additionally, we predicted expected parasitism rate, which captures changes in attack rates, but not subsequent changes in host population abundances; a prerequisite for demonstrating apparent competition between host species mediated by shared parasitoids. However, previous work on the same study system showed that accurate predictions of changes in host abundances followed from the ability to predict changes in attack rates, and were, in fact, better predicted than changes in attack rates [21].

## Further caveats

Ecological networks are difficult to sample and the difficulty of distinguishing true non-interactions from unobserved interactions, which may simply result from inadequate sampling, is widely recognised [56, 101–103]. Although the 'meta-networks' used to train our machine-learning models (see 'Training meta-networks' in S1 Appendix) comprised the pooled data across training sites and sampling dates, to maximise the resolution of potential host-parasitoid links, we acknowledge that these meta-networks may not capture all realised interactions.

A further caveat to using these approaches in biocontrol is that predicting interactions involving exotic species (e.g., control agents) may be more difficult than predicting interactions among native species. The lack of co-evolutionary history between exotic and native species in the community suggests that trait or phylogenetic matching will be weaker for these interactions. In fact, Peralta *et al*. [59] found that niche processes were weaker, and Coux *et al*. [60] found that neutral processes were stronger, for plant-frugivore interactions involving exotic species than those among native species, suggesting that methods incorporating trait or phylogenetic matching will perform poorly for interactions involving exotic species compared to interactions between native species. For this reason, an approach that blends information on traits and abundances (as we did in the second linear modelling step) may help to alleviate the problems of predicting more neutrally-driven interactions. In addition, exotic species tend to interact with numerous species [50, 104, 105] and, as discussed above, interactions involving generalists may be more difficult to predict based on trait matching than those involving specialists [59]. However, this is less of a concern as biocontrol agents are typically chosen because they are specialists on the target pest.

## Conclusion

Previous studies show that biocontrol agents can have negative non-target impacts, and stress the importance of considering these effects in biocontrol risk assessment [84]. Our findings show that machine-learning methods offer a promising approach to begin to predict direct effects, and to a lesser extent, indirect effects, using information on species traits, abundances and phylogenies. In addition, the ability to predict species interactions, and more specifically non-target impacts of biocontrol agents, was hypothesised to vary with species characteristics and habitat type [59]. Congruent with the first of these assertions, we found that overall

predictions of interaction frequencies (i.e. direct effects) varied with the generality and abundance of the interaction partners (host and parasitoid). However, predictive ability did not depend on the habitat type of the training data. This supports the use of a common dataset of species interactions and traits for risk assessment.

## Supporting information

**S1 Appendix. Trait and phylogenetic details, machine-learning model details, hyperparameter tuning details, supplementary analysis and coefficient tables for analyses.**
(DOCX)

**S2 Appendix. Host-parasitoid interaction data (test sites).** Note: N = native forest, P = plantation forest, EH = host reduction treatment, EC = control, extra = whether the caterpillar was collected during extra sampling, B = before host-reduction treatment (i.e. at time *t*), A = after host-reduction treatment (i.e. at time *t+1*).
(CSV)

**S3 Appendix. Host body size data.**
(CSV)

**S4 Appendix. Parasitoid body size data.**
(CSV)

**S5 Appendix. GenBank accession numbers for host species.**
(CSV)

**S6 Appendix. GenBank accession numbers for parasitoid species.**
(CSV)

**S7 Appendix. Host-parasitoid interaction data (training sites).**
(CSV)

**S1 Fig.**
(TIF)

## Acknowledgments

We thank Anna Eklöf and Alex James for helpful comments on an earlier draft and Barbara Barratt, Rogini Runghen and Pieter Pelser for helpful discussions.

Our code is available on GitHub (https://github.com/hannah-jk/predicting_nontarget_impacts).

## Author Contributions

**Conceptualization:** Jacqui H. Todd, Jason M. Tylianakis.

**Formal analysis:** Hannah J. Kotula.

**Investigation:** Guadalupe Peralta, Carol M. Frost.

**Methodology:** Hannah J. Kotula, Guadalupe Peralta, Carol M. Frost, Jason M. Tylianakis.

**Supervision:** Guadalupe Peralta, Jason M. Tylianakis.

**Writing – original draft:** Hannah J. Kotula.

**Writing – review & editing:** Hannah J. Kotula, Guadalupe Peralta, Carol M. Frost, Jacqui H. Todd, Jason M. Tylianakis.

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
