## [Decision Letter · Decision Letter 0]

22 Dec 2020

PONE-D-20-35961

Predicting direct and indirect non-target impacts of biocontrol agents using machine-learning approaches

PLOS ONE

Dear Dr. Kotula,

Thank you for submitting your manuscript to PLOS ONE. After careful consideration, we feel that it has merit but does not fully meet PLOS ONE’s publication criteria as it currently stands. Therefore, we invite you to submit a revised version of the manuscript that addresses the points raised during the review process.

Both reviewers found the topic of this work interesting and the approach innovative. However both reviewers also found the manuscript hard to follow, with reviewer one commenting "This paper was quite heavy to work through" and reviewer also remarking on difficulties following the text and suggesting that a conceptual diagram to better guide readers through the work might be helpful. Overall, this manuscript needs to be heavily revised for clarity and readability. Both reviewers also list numerous additional specific issues that need to be addressed. It will be essential that you respond directly to all comments in a revision.

We look forward to receiving your revised manuscript.

Kind regards,

Patrick R Stephens, Ph.D.

Academic Editor

PLOS ONE

Journal Requirements:

2.) Thank you for stating the following in the Competing Interests section:

'The authors have declared that no competing interests exist.'

We note that one or more of the authors are employed by a commercial company: The New Zealand Institute for Plant and Food Research Limited.

a.) Please provide an amended Funding Statement declaring this commercial affiliation, as well as a statement regarding the Role of Funders in your study. If the funding organization did not play a role in the study design, data collection and analysis, decision to publish, or preparation of the manuscript and only provided financial support in the form of authors' salaries and/or research materials, please review your statements relating to the author contributions, and ensure you have specifically and accurately indicated the role(s) that these authors had in your study. You can update author roles in the Author Contributions section of the online submission form.

b.) Please also provide an updated Competing Interests Statement declaring this commercial affiliation along with any other relevant declarations relating to employment, consultancy, patents, products in development, or marketed products, etc.  

Reviewers' comments:

Reviewer's Responses to Questions

**Comments to the Author**

1. Is the manuscript technically sound, and do the data support the conclusions?

Reviewer #1: Partly

Reviewer #2: Yes

2. Has the statistical analysis been performed appropriately and rigorously? 

Reviewer #1: I Don't Know

Reviewer #2: Yes

3. Have the authors made all data underlying the findings in their manuscript fully available?

Reviewer #1: Yes

Reviewer #2: Yes

4. Is the manuscript presented in an intelligible fashion and written in standard English?

Reviewer #1: Yes

Reviewer #2: Yes

5. Review Comments to the Author

Reviewer #1: This is a review of the manuscript entitled "Predicting direct and indirect non-target impacts of biocontrol agents using machine-learning approaches". Overall, I found the manuscript quite interesting, as link prediction approaches in complex networks is a focus of my own work. However, the manuscript was incredibly confusing at many points, which made it slightly less enjoyable to read. Despite this, the authors find some interesting results. I think some of the main results about predictive performance should be tempered, as the overall model performance was actually quite poor, but given the complexities of host-parasitoid interactions and the difficulty in starting to address indirect interactions, this is still quite interesting. I hope my comments are constructive, and are not simply from my misunderstandings. This paper was quite heavy to work through.

## General comments

The models that predict direct interactions in these networks aren't actually predicting interactions, but identifying potentially suitable interactions. This seems like nuance, but it feels like nuance that needs to be discussed further in the manuscript. Especially when only considering trait relationships with a limited number of covariates, it's easy to imagine a situation where a model would predict a high interaction suitability, ignoring some aspect of biological realism (different salt tolerance, spatial/temporal niche partitioning, etc. etc.).

Interaction frequency isn't _really_ the same as probability of an interaction though, right? (line 217-220). Interaction frequency is likely driven by abundance of the interactors, and deals with the activity of the interactors. The probability of interaction driven by traits is something different. I think this point could be clarified, along with the differences from other link prediction work, and a bit more effort could distinguish the direct host-parasitoid trait matching models with the subsequent time-lag predicted attack rate work. I think this will be a main source of confusion to the reader, though this is based on my own experience as a reader.

How was phenology measured? Introduced around line 200, but not defined in main text. Perhaps add in a quick one sentence (it's abundance per month, I believe?) and then reference the Appendix 1? Getting as much information into the main text in a clearer way will really help the reader here, since so much of the relevant detail is in the set of 6 appendices.

The test for indirect interactions seems strange. It is a regression of observed parasitism rates at t+1 as a function of expected parasitism rate and site (random effect). But this is more of a model validation estimation than a detection of indirect effects, unless I'm misunderstanding. For instance, no indirect effects could be present in the data, and a well-trained model capable of predicting parasitism rate would still show a clear relationship to the observed parasitism rates. I think I must be missing something. There is some discussion of the validity of this on lines 323-332, but wouldn't the model still fit well if there were no indirect interactions but instead the parasite transmission was simply density dependent. I guess the argument is that reductions in host abundance that result in reductions in parasite abundance form the basis of an indirect interaction iff the attack rate on another host is reduced...which it would be as a function of that host's density if it was also reduced. I'm having trouble wrapping my head around how indirect interactions are being quantified and if this really consitutes an indirect interaction, or simply the result of density-dependent transmission and a perturbed host community.

The biocontrol angle seems a bit strange, but that's the authors' choice and it's fine. I think maybe I was just dense, but I kept thinking of biocontrol as the introduction of a species to control a parasitoid, and I struggled to rationalize that with the KNN approach as such, since it assumes the host and parasitoid communities and their traits are known quantities. I think the authors are arguing that using pesticide to suppress the host community (or specific members therein) is biocontrol. If this is clear to others, ignore this. Otherwise, perhaps be very clear about this in the introductory text? The idea is that the suppression of the host community can shift host abundances in a way to promote indirect effects of hosts via shared parasitoids. Correct me if I'm wrong on this. This paper is great, but it is quite confusing and dense.

Please consider making the code available to work through the data, as the current data citation links to a previous study on phylogenetic diversity patterns at habitat edge.

## Specific comments

line 87: consider changing "pure" to "basic" or something similar (e.g., 'theoretical'). The word "pure" has somewhat of an odd and subjective feel to it.

line 90: provide a couple citations after "...frame has been widely used in ecology..."

line 115: known interactions and a set of background interactions to get a sense of the available trait space, right? Training on known interactions would be a regression where the response variable has no variation.

line 122: Consider re-wording slightly. Species abundances are related to species traits, such that using data on species abundances bakes in some trait variation, and doesn't really reflect neutral processes, arguably.

line 593: is an R2 of around 0.1 "reasonable"? If the authors permuted their predictor variables and re-ran all the models, what would the resulting R2 be? It's possible that random variation and odd model behavior could result in an R2 comparable to that from an entirely uninformed model.

figures: it would be a bit nicer if the figure captions and the figures were together. This could be a journal submission issue though. Resolution on the figures also seems a bit off to me.

Figure 1: It looks like there were only ~ 7 different values the observed interaction frequency values took? And those lines of best fit are not very good (seems like many of the assumptions of the linear model would be violated here).

I'm not sure how much Figure 2 really shows the impact of degree (generality). Is there a way to maybe add an alpha channel to the black points, as I imagine many of the points on plotted on top of others here?

So many of the quantities are scaled in the figures, and I'm not clear on the point of this. I sort of wanted to see the actual numbers for things like expected parasitism rate. If scaling simply centers the data such that it's still proportional to the original, then show the true data. The use of the plotting the residuals also was a bit confusing, but I believe I understand the utility there.

Reviewer #2: Dear Dr. Stephens, dear authors

The manuscript ‘predicting direct and indirect non-target impacts of biocontrol agents using machine-learning approaches’ by Kotula et al. present a new approach to predict indirect effects of biocontrol agents by using predictions for direct interactions informed by machine-learning (ML) algorithms.

They do so by first training and evaluating ML algorithms (Random Forest and kNN) on sites with observed host-parasitoid interactions at time t, and using the predicted interaction outcomes to predict indirect effects (apparent competition) at time t+1 after an artificially removal of hosts. The authors generally found that RF predicts direct interactions better than kNN and that while the species generality influences the predictive performance, habitat type surprisingly does not. They also found no difference in predictive performance when using ML predictions or data-based based values when inferring the indirect effects, demonstrating the potential of ML here since data about species interactions are scarce. However, the overall predictive performance for indirect effects is low.

General Comments ==========

I found the general idea to use ML to infer the consequences of biocontrol agents on the community very appealing and suitable for PLOS ONE. We are all aware of the difficulty of sampling species interactions and especially the difficulty and cost of a controlled experiment with a biocontrol agent. ML could provide a great opportunity here to calculate different scenarios theoretically, as long as the validity of ML algorithms for this approach is ensured.

The authors showed that they are very well versed in ML and statistic: a) correct validation setup (i.e. spatially blocked CV), b) hyper-parameter tuning (even RF requires hyper-parameter tuning), c) accounting for neutral processes (species abundances) by using them as predictors, d) exploring the impact of habitat types on the predictive performance, e) correcting p-values for multiple testing, and f) stressing the problem of model-selection. In particular, c) the point that they accounted for species abundances was very important because previous studies showed that they are alone could be important predictors of species interactions and they indeed showed here high variables importances.

In summary, I was very convinced by the authors’ work and therefore have only one major and few minor comments.

==== Major Comments ====

I found it difficult to follow the overall approach, the authors first explore the predictive performance for the species interactions (+ exploration of different predictors), then they explore the predictive performance for the indirect effects (based on the previous predictions and the true observations), and the evaluation of the different predictions is done using mixed effect models, for which they did a model selection. All of this requires the reader to process a lot of information, and I often found myself longing for a conceptional figure of the approach. In particular, when reading through the results and the discussion it took me a while to understand everything just from the text. I believe that conceptional figure of the approach would be very helpful to the audience and would also make the work more accessible for the community.

==== Minor Comments ====

Regarding species abundances, could you please provide the actual variable importances of the RF when predicting the species interactions? You wrote that the species abundances were the most importance predictors, but I would like to see them compared to the other predictors. If they are twice as importance or even more important, then I suspect that you could improve the overall predictive performance by removing them as predictors but correcting the observed species interactions by the abundances (to force the model to learn patterns that generalize better?), or you could include them as transformed weights into the model (some random forest implementations have options for observational weights (e.g. ranger)).

As for the hyper-parameter tuning, you did not use here a blocked CV setup, right? Blocking species (host or parasitoids, or even both) might improve the predictive performance for the direct and also indirect effects, since the current CV introduces dependencies between the species for which the tuning is optimized for. Also, using a kernel might improve the performance for the kNN.

Regarding the mixed-effect models, why do you use model-selection? As I understand it, you used mixed-effect models for two reasons: a) to evaluate the predictive performance (in the simplest case if predictions and observations are on the same scale, we would expect an effect of 1.0 for the observations) and b) to disentangle the different effects such as site, generality of species, and so on. However, for a) it would be fine to use model selection since you are only interested in the total predictive performance that can be achieved, but for b) you are interested in the explanatory model and here model-selection makes no sense because model-selection with AIC does not select the ‘true’ (‘correct’ model), but only the best predictive model. So, I suggest that you either use different models for the two questions or use commonly known accuracy measurements for a) without using a mixed effect model.

For reproducibility, but also for the community to use your approach, I find it indispensable to provide all the necessary code to reproduce the analysis/method. Do you plan to upload your code on a freely accessible platform such as GitHub?

To improve the accessibility of the manuscript, I suggest that you add the formulas (R formulas) for all mixed effect models.

Line specific comments ==========

L27: There is already much information in the abstract, consider removing it

L28: machine-learning, consistency!

L39: I would like to see a stronger conclusion here

L39: I am not exactly sure what you mean here with 'explanatory power'? The predictions of ml is a weak effect in the final predictive model for the indirect interactions?

L130-131: Yes, see also Poisot et al 2015 (10.1111/oik.01719)

L135: See Poisot et al 2015, you need distinguish yourself from this work (which is no problem but you have to show that you are well informed about the relevant literature)

L138: in the first line of the introduction you write 'nontarget' and here 'non-target', consistency!

L143-144: I assume that this is indeed possible, but to identify the (predictive) underlying rules which generalizes well over scales/habitat types would require many different datasets from different scales to control for all the scale effects and force the ML model to identify 'global' patterns.

L272: which feature importance? Gini?

L277: interesting, previous works reported that phylogenic predictors are important.

L281: I wonder if it wouldn't be better to correct the interaction outcomes by the abundances to force the models to learn other predictive rules. I fully agree with you that abundance can be a very important predictor but this is exactly the problem, because the model could mainly use abundance and other correlative predictors could be neglected, leading to a less generalisable model if in the next dataset the abundance pattern is different.

Please report the individual importances

L346: I found this section difficult to follow, it might help the reader to give information about the regression models with the commonly known formula syntax (R, lme4)

L370-375: Excellent!

L384: Why?

L423: Again, why?

L457: Concistency? Why use previously AIC and now AICc? Could you please also provide more information about the dimensions of the data? E.g. how many observations did you have for the different habitat types?

L523-525: Here, you interpreted the model causally, but you selected for the best predictive model via AIC

L581-584: I aggree, finding a correct threshold for new data is very difficult/non-trivial

6. PLOS authors have the option to publish the peer review history of their article (what does this mean?). If published, this will include your full peer review and any attached files.

Reviewer #1: No

Reviewer #2: No

---

## [Author Response · Author response to Decision Letter 0]

26 Feb 2021

Kotula et al. Response to Reviewers

26 February 2021

Dear Dr. Stephens,

My co-authors and I would like to thank you and the two anonymous reviewers for the positive and constructive comments on our manuscript. We appreciate the suggestions, and I have consulted with all co-authors on revisions, which we present in point-by-point responses (in purple text) to each comment below, along with tracked changes in the submitted manuscript and supporting information. We feel that the reviewers’ comments have been addressed appropriately, and that the paper is now improved as a result. In summary, following the reviewers’ suggestions, we have made a change to our analyses (in particular, we have removed abundance from our machine-learning models). Our results have remained similar following this change. Additionally, we have included a summary diagram outlining our methods, and have revised our manuscript for clarity. If you have any further questions, concerns, or revisions, please do not hesitate to contact us.

Kind regards,

Hannah Kotula, Guadalupe Peralta, Carol Frost, Jacqui Todd, and Jason Tylianakis 

Amended funding statement:

H.J.K acknowledges support from a Roland Stead Postgraduate Scholarship in Biology. HJK, GP and JMT were funded by the Marsden Fund (grant number UOC1705), administered by the Royal Society of New Zealand. J.H.T is funded by the Better Border Biosecurity (B3) (www.b3nz.org) research collaboration funded by the New Zealand Government.

We acknowledge that one of the authors (J.H.T) is employed by a commercial company (The New Zealand Institute for Plant and Food Research Limited). The funder did not provide support in the form of salaries for an author [J.H.T], and did not have any additional role in the study design, data collection and analysis, decision to publish, or preparation of the manuscript. The specific role of this author is articulated in the ‘author contributions’ section. 

Amended competing interests statement:

The authors have declared that no competing interests exist. We acknowledge that one of the authors (J.H.T) is employed by a commercial company (The New Zealand Institute for Plant and Food Research Limited). This does not alter our adherence to PLOS ONE policies on sharing data and materials. 

Editor comments:

Thank you for submitting your manuscript to PLOS ONE. After careful consideration, we feel that it has merit but does not fully meet PLOS ONE’s publication criteria as it currently stands. Therefore, we invite you to submit a revised version of the manuscript that addresses the points raised during the review process.

Both reviewers found the topic of this work interesting and the approach innovative. However both reviewers also found the manuscript hard to follow, with reviewer one commenting "This paper was quite heavy to work through" and reviewer also remarking on difficulties following the text and suggesting that a conceptual diagram to better guide readers through the work might be helpful. Overall, this manuscript needs to be heavily revised for clarity and readability. Both reviewers also list numerous additional specific issues that need to be addressed. It will be essential that you respond directly to all comments in a revision.

We appreciate all of the suggestions given and present our revisions in point-by-point responses (in purple text) to each comment below, as well as tracked changes in the submitted manuscript. We have followed all the specific recommendations of the reviewers, including addition of a figure summarising our methods, and have revised our manuscript for clarity. 

Reviewer #1: This is a review of the manuscript entitled "Predicting direct and indirect non-target impacts of biocontrol agents using machine-learning approaches". Overall, I found the manuscript quite interesting, as link prediction approaches in complex networks is a focus of my own work. However, the manuscript was incredibly confusing at many points, which made it slightly less enjoyable to read. Despite this, the authors find some interesting results. I think some of the main results about predictive performance should be tempered, as the overall model performance was actually quite poor, but given the complexities of host-parasitoid interactions and the difficulty in starting to address indirect interactions, this is still quite interesting. I hope my comments are constructive, and are not simply from my misunderstandings. This paper was quite heavy to work through.

Thanks for your comments. To increase the clarity of our manuscript, we have now included a conceptual diagram summarising the methods, and where possible we have changed wording to increase clarity.

The second reviewer suggested a change to the analyses (specifically, removing abundance from the machine-learning methods). We have made this change and re-run the analyses. The predictive ability of our machine-learning methods for direct effects has now improved, though predictive ability of indirect effects has remained similar. We agree that our predictive ability of indirect effects is not very high, and acknowledge this in both the abstract (L40-42), and discussion (L844-845). (Note that line numbers are for the ‘Revised Manuscript with Track Changes’). 

## General comments

The models that predict direct interactions in these networks aren't actually predicting interactions, but identifying potentially suitable interactions. This seems like nuance, but it feels like nuance that needs to be discussed further in the manuscript. Especially when only considering trait relationships with a limited number of covariates, it's easy to imagine a situation where a model would predict a high interaction suitability, ignoring some aspect of biological realism (different salt tolerance, spatial/temporal niche partitioning, etc. etc.).

We acknowledge that our machine-learning models likely do not capture all aspects of biological realism. However, previous work has shown that relatively few traits may be needed to explain network structure, so we now mention both of these points (L115-118):

“In addition to traits being useful predictors of interactions, sometimes relatively few traits are needed to explain network structure (Eklöf et al. 2013), though in all cases these predictions are of potentially suitable interactions given the imperfect knowledge of traits and potential for co-occurrence to limit the realisation of these potential interactions.”

Nevertheless, we selected traits that have been identified as being useful for predicting interactions (L202-203). We accounted for spatial and temporal partitioning by only predicting interactions for host-parasitoid species pairs that co-occurred at a site (L298-300), and by including measures of both host and parasitoid phenology as predictors in our machine-learning models (L212-215). We also included phylogenies as a surrogate for other potentially unmeasured traits.

Interaction frequency isn't _really_ the same as probability of an interaction though, right? (line 217-220). Interaction frequency is likely driven by abundance of the interactors, and deals with the activity of the interactors. The probability of interaction driven by traits is something different. I think this point could be clarified, along with the differences from other link prediction work, and a bit more effort could distinguish the direct host-parasitoid trait matching models with the subsequent time-lag predicted attack rate work. I think this will be a main source of confusion to the reader, though this is based on my own experience as a reader.

We agree that frequency and probability aren’t identical, though one of our machine-learning methods (KNN) predicts the frequency rather than the probability of that frequency. We nevertheless show that higher interaction probabilities correspond with higher observed interaction frequencies (L563-565), suggesting these quantities are related.

We also agree that niche processes may determine which interactions are possible, whereas abundance may determine the frequency (or even realisation) of interactions. Reviewer 2 makes a similar comment, and we have now removed abundance from the random forest and KNN models (such that these model now only capture niche processes) and have added abundance later (in the linear mixed models predicting interaction frequency and occurrence) to account for the influence of abundance on interaction frequency, given a niche-based prediction. We have also added to the introduction (L130-132) the sentence: “While species traits may determine which interactions are possible (e.g., ‘forbidden links’; Jordano et al. 2003), the occurrence frequency of interactions may be driven by species abundances (Stang et al. 2007),” to make this clear from the start. 

One of the main distinguishing features of our work is the indirect effect predictions, which we introduce in the introduction.

We have now included a conceptual diagram summarising our methods to both increase the clarity of our research and help distinguish between the prediction of direct and indirect effects. 

How was phenology measured? Introduced around line 200, but not defined in main text. Perhaps add in a quick one sentence (it's abundance per month, I believe?) and then reference the Appendix 1? Getting as much information into the main text in a clearer way will really help the reader here, since so much of the relevant detail is in the set of 6 appendices.

Thank you for this suggestion. We have added one sentence (L212-215) explaining how phenology was measured and have referenced the relevant section in Appendix S1.

The test for indirect interactions seems strange. It is a regression of observed parasitism rates at t+1 as a function of expected parasitism rate and site (random effect). But this is more of a model validation estimation than a detection of indirect effects, unless I'm misunderstanding. For instance, no indirect effects could be present in the data, and a well-trained model capable of predicting parasitism rate would still show a clear relationship to the observed parasitism rates. I think I must be missing something. There is some discussion of the validity of this on lines 323-332, but wouldn't the model still fit well if there were no indirect interactions but instead the parasite transmission was simply density dependent. I guess the argument is that reductions in host abundance that result in reductions in parasite abundance form the basis of an indirect interaction iff the attack rate on another host is reduced...which it would be as a function of that host's density if it was also reduced. I'm having trouble wrapping my head around how indirect interactions are being quantified and if this really consitutes an indirect interaction, or simply the result of density-dependent transmission and a perturbed host community.

Thank you for the opportunity to clarify. We specifically test for apparent competition (an indirect effect), and quantify apparent competition using Equation 6 from Müller et al. (1999) (as we describe on L349). To summarise our approach we have added (L329-338):

“To predict indirect interactions (apparent competition), we used the quantitative interaction network (i.e. direct interactions at each test site at time t) predicted by each model (random forest or KNN) trained on the ‘combined’ data (native forest and plantation data). This quantitative interaction network can be used to project a network of indirect linkages among hosts that share parasitoids. We can then predict how attack rates on each host should change as a function of changes in the abundance of host species that share parasitoids with that host (i.e. the indirect effect of ‘apparent competition’). Therefore, we predicted how patterns in attack would change if the indirect (parasitoid-sharing) links among hosts matched those predicted by the machine-learning methods. We then went on to validate how these predicted changes in attack rate matched real changes observed (between times t and t+1).” 

As validation of the overall approach, previous work on the same study system found that indirect effects were present in this system and detectable using this approach (Frost et al. 2016). We have now added this point on L161. If there were no indirect interactions, then the predictions of changing attack rates would be poor, because these predictions are based on proportional parasitoid sharing among hosts, rather than direct host-parasitoid interaction rates. In support of this argument, Frost et al. (2016) found that excluding within-habitat intraspecific contributions (that is, delayed density dependent parasitism) from the calculation of expected parasitism rate did not significantly change the ability of expected parasitism rate to predict observed parasitism rate. Therefore, the indirect effects present cannot simply be explained by within-habitat delayed density dependence. We have also added this point (L378-382). 

The biocontrol angle seems a bit strange, but that's the authors' choice and it's fine. I think maybe I was just dense, but I kept thinking of biocontrol as the introduction of a species to control a parasitoid, and I struggled to rationalize that with the KNN approach as such, since it assumes the host and parasitoid communities and their traits are known quantities. I think the authors are arguing that using pesticide to suppress the host community (or specific members therein) is biocontrol. If this is clear to others, ignore this. Otherwise, perhaps be very clear about this in the introductory text? The idea is that the suppression of the host community can shift host abundances in a way to promote indirect effects of hosts via shared parasitoids. Correct me if I'm wrong on this. This paper is great, but it is quite confusing and dense.

As the reviewer suggests, classical biocontrol is the introduction of novel species to control pest species, and our manuscript is most relevant to parasitoid biocontrol agents introduced for the control of pest host species (as we mention in the introduction: L62-63 and L148-149). Also, we explain in L158 that: “…rather than focusing on biocontrol species only, we focus on developing a proof of concept using an entire host-parasitoid community (only some of which are pests and biocontrol agents) …”. In terms of the reviewer’s point that “it assumes the host and parasitoid communities and their traits are known quantities”, this is correct, and also true of real-world biocontrol. A target community (including one or more pest species) occurs in a location, and can be measured. Then when a putative enemy is selected, its traits can also be measured, and its interactions with the target community can be predicted using the approaches we present here.

In the introduction we outline the data requirements for both random forest and KNN (L148-154), and we further explain them in the ‘Comparison of data requirements for random forest and KNN’ section in Appendix S1, so hopefully it’s clear that we assume these to be known quantities.

The pesticide treatment was only to generate changes in abundance to test our predictions – the pesticide isn’t part of biocontrol practice.

Yes, shared parasitoid species may mediate indirect effects among host species, as we explain in L62: “In particular, populations of herbivore species may be linked via their shared natural enemies (e.g. parasitoids, which are often used as biocontrol agents), such that an increase in one host herbivore population may drive an increase in the abundance of a shared enemy and result in a decrease in another host’s population (i.e., apparent competition; Holt 1977). In this way, parasitoid species may mediate indirect effects, such as apparent competition, among host species.”

We recognise that the manuscript presents a lot of information, and that the methodological part can make it dense. To make the manuscript more clear, we’ve added the diagram (Fig 1) in which we show the different steps of the methodology and analyses we performed. 

Please consider making the code available to work through the data, as the current data citation links to a previous study on phylogenetic diversity patterns at habitat edge.

Our code will be made available on GitHub upon manuscript acceptance. 

## Specific comments

line 87: consider changing "pure" to "basic" or something similar (e.g., 'theoretical'). The word "pure" has somewhat of an odd and subjective feel to it.

We’ve changed it to ‘theoretical’. 

line 90: provide a couple citations after "...frame has been widely used in ecology..."

We’ve added three citations. 

line 115: known interactions and a set of background interactions to get a sense of the available trait space, right? Training on known interactions would be a regression where the response variable has no variation.

In the introduction we explain (starting L121): “These algorithms are trained using known interactions, which allow the machine-learning technique to learn which trait and/or phylogenetic combinations are likely to correspond to an interaction occurrence, and these can then be used to predict interactions among new species.”

We trained the machine-learning models (random forest and KNN) on known interactions (along with trait and phylogenetic data), so in this case we’re not using a regression. 

line 122: Consider re-wording slightly. Species abundances are related to species traits, such that using data on species abundances bakes in some trait variation, and doesn't really reflect neutral processes, arguably.

We agree and we have added the following sentence (L128): “Although ecological and life history traits can influence species abundances (Bartomeus et al. 2016), abundance is usually considered to also result from neutral processes (Volkov et al. 2003, Krishna et al. 2008, Vázquez et al. 2009, Canard et al. 2014).”

line 593: is an R2 of around 0.1 "reasonable"? If the authors permuted their predictor variables and re-ran all the models, what would the resulting R2 be? It's possible that random variation and odd model behavior could result in an R2 comparable to that from an entirely uninformed model.

We have changed the wording and this line (starting 725) now reads: 

“Although these models did not explain a large proportion of the variance (random-forest marginal R2 =0.155, KNN-marginal R2 = 0.0919), in nature we would expect many processes beyond just trait matching to determine field parasitism rates, though we take the significant effects of both RF- and KNN-predicted parasitism rates as evidence that these variables were informative.”

Additionally, in the discussion we acknowledge that (L844-845): “These methods are therefore a long way from being able to make accurate predictions of the full strength of parasitoid-mediated indirect effects.” 

We use model selection to determine which predictors should be retained in the best-fitting model, and in these models the predictor variable (expected parasitism rate) was retained, so we would conclude that the models are not entirely uninformative. 

figures: it would be a bit nicer if the figure captions and the figures were together. This could be a journal submission issue though. Resolution on the figures also seems a bit off to me.

The journal requires the figures and legends to be separate. We’ve increased the resolution of the figures. 

Figure 1: It looks like there were only ~ 7 different values the observed interaction frequency values took? And those lines of best fit are not very good (seems like many of the assumptions of the linear model would be violated here).

Yes, there were 5 different values of observed interaction frequency for host-parasitoid species pairs at the first-time step (t) test sites, ranging from 0 to 11. Note that figure 2 (we have added a figure so it’s now figure 2) shows the residuals on the y axis, and therefore has a different range of values. 

Figure 2 shows the results from Poisson linear mixed models. We have add “tested using Poisson linear mixed models” to the figure caption to make this more clear. We checked the assumptions of all models and they were met in all cases (as we explain on L543). 

We apologise that we mistakenly had written that each point in the figure represents a host species. We have corrected this to say that each point represents a host-parasitoid species pair (in the caption for figure 2). 

I'm not sure how much Figure 2 really shows the impact of degree (generality). Is there a way to maybe add an alpha channel to the black points, as I imagine many of the points on plotted on top of others here?

In the figure caption we explain how predictions of observed interaction frequency vary with the generality of the interacting partners, and we have added a few more words to make it more clear (L673). 

Thanks for the suggestion – we have now used ‘jitter’ to better see overlapping points. 

So many of the quantities are scaled in the figures, and I'm not clear on the point of this. I sort of wanted to see the actual numbers for things like expected parasitism rate. If scaling simply centers the data such that it's still proportional to the original, then show the true data. The use of the plotting the residuals also was a bit confusing, but I believe I understand the utility there.

We have included an additional sentence explaining that scaling both centres and scales the predictor variable such that it has a mean of 0 and a standard deviation of 1, and we explain why we scaled predictor variables in L413-414. Because we scaled predictor variables for all models, we used scaled predictors in the figures, to ensure that the figures accurately represent the model results, which form the basis of our interpretation. 

Reviewer #2: Dear Dr. Stephens, dear authors

The manuscript ‘predicting direct and indirect non-target impacts of biocontrol agents using machine-learning approaches’ by Kotula et al. present a new approach to predict indirect effects of biocontrol agents by using predictions for direct interactions informed by machine-learning (ML) algorithms.

They do so by first training and evaluating ML algorithms (Random Forest and kNN) on sites with observed host-parasitoid interactions at time t, and using the predicted interaction outcomes to predict indirect effects (apparent competition) at time t+1 after an artificially removal of hosts. The authors generally found that RF predicts direct interactions better than kNN and that while the species generality influences the predictive performance, habitat type surprisingly does not. They also found no difference in predictive performance when using ML predictions or data-based based values when inferring the indirect effects, demonstrating the potential of ML here since data about species interactions are scarce. However, the overall predictive performance for indirect effects is low.

General Comments ==========

I found the general idea to use ML to infer the consequences of biocontrol agents on the community very appealing and suitable for PLOS ONE. We are all aware of the difficulty of sampling species interactions and especially the difficulty and cost of a controlled experiment with a biocontrol agent. ML could provide a great opportunity here to calculate different scenarios theoretically, as long as the validity of ML algorithms for this approach is ensured.

The authors showed that they are very well versed in ML and statistic: a) correct validation setup (i.e. spatially blocked CV), b) hyper-parameter tuning (even RF requires hyper-parameter tuning), c) accounting for neutral processes (species abundances) by using them as predictors, d) exploring the impact of habitat types on the predictive performance, e) correcting p-values for multiple testing, and f) stressing the problem of model-selection. In particular, c) the point that they accounted for species abundances was very important because previous studies showed that they are alone could be important predictors of species interactions and they indeed showed here high variables importances.

In summary, I was very convinced by the authors’ work and therefore have only one major and few minor comments.

==== Major Comments ====

I found it difficult to follow the overall approach, the authors first explore the predictive performance for the species interactions (+ exploration of different predictors), then they explore the predictive performance for the indirect effects (based on the previous predictions and the true observations), and the evaluation of the different predictions is done using mixed effect models, for which they did a model selection. All of this requires the reader to process a lot of information, and I often found myself longing for a conceptional figure of the approach. In particular, when reading through the results and the discussion it took me a while to understand everything just from the text. I believe that conceptional figure of the approach would be very helpful to the audience and would also make the work more accessible for the community.

We thank the reviewer for their comments, and have included a diagram (Figure 1) summarising the methods to increase the clarity of our research. 

==== Minor Comments ====

Regarding species abundances, could you please provide the actual variable importances of the RF when predicting the species interactions? You wrote that the species abundances were the most importance predictors, but I would like to see them compared to the other predictors. If they are twice as importance or even more important, then I suspect that you could improve the overall predictive performance by removing them as predictors but correcting the observed species interactions by the abundances (to force the model to learn patterns that generalize better?), or you could include them as transformed weights into the model (some random forest implementations have options for observational weights (e.g. ranger)).

Thanks for the suggestion. Parasitoid abundance was about three times as important as the second most important parasitoid trait, and host abundance was the fourth most important host trait (Table 20 in Appendix S1). Following the reviewer’s suggestion, we have now removed host and parasitoid abundance from both the random-forest and KNN models, and you were right that this improved predictions of interaction frequencies. We provide variable importances for our random-forest models in the appendix (and have updated these values for the new set of traits excluding abundance) and have added “(Table 20 in Appendix S1)” to L323. We now include abundance separately in the subsequent step of prediction (our linear models), so as to account for its influence without interfering with the trait-based predictions.

As for the hyper-parameter tuning, you did not use here a blocked CV setup, right? Blocking species (host or parasitoids, or even both) might improve the predictive performance for the direct and also indirect effects, since the current CV introduces dependencies between the species for which the tuning is optimized for. Also, using a kernel might improve the performance for the kNN.

We have now blocked by parasitoid species. We chose to block by parasitoids rather than by hosts because in a biocontrol context host species may be shared between locations, whereas existing interaction data for a parasitoid control agent (in its home range for example) may not be available. Also, in our study system there is overlap in host species between the training and testing sites. Note that we would not be able to block by both host and parasitoid species because our training networks (which we used for hyperparameter tuning) don’t contain separate groups of parasitoid species that interact with non-overlapping subsets of host species. Also, note that we use the term ‘group’ rather than ‘block’ in the manuscript to be consistent with the name of the Python function GroupKFold() which we used to group by parasitoid species.

For the KNN parameter search, we included runs with both weighted and unweighted neighbours (‘Tuning hyperparameters for KNN’ in Appendix S1), and found that using weighted neighbours was better. We therefore used weighted neighbours for predicting interactions at the test sites). We explain the weighting function we used in ‘Tuning hyperparameters for KNN’ in Appendix S1. 

Regarding the mixed-effect models, why do you use model-selection? As I understand it, you used mixed-effect models for two reasons: a) to evaluate the predictive performance (in the simplest case if predictions and observations are on the same scale, we would expect an effect of 1.0 for the observations) and b) to disentangle the different effects such as site, generality of species, and so on. However, for a) it would be fine to use model selection since you are only interested in the total predictive performance that can be achieved, but for b) you are interested in the explanatory model and here model-selection makes no sense because model-selection with AIC does not select the ‘true’ (‘correct’ model), but only the best predictive model. So, I suggest that you either use different models for the two questions or use commonly known accuracy measurements for a) without using a mixed effect model.

Although we agree that model-selection with AIC may not select the true model, our focus is prediction of observed interaction frequencies (rather than explaining the true causes of interactions). Also, we know from other research that interactions can differ among habitats, and can depend on species generality and abundance (L138); our objective here is to see if we need to include this information into predictive models to make them accurate, or whether the influence of these variables can be ignored and reasonable predictions can still be made. 

For reproducibility, but also for the community to use your approach, I find it indispensable to provide all the necessary code to reproduce the analysis/method. Do you plan to upload your code on a freely accessible platform such as GitHub?

Our code will be made available on GitHub upon manuscript acceptance.

To improve the accessibility of the manuscript, I suggest that you add the formulas (R formulas) for all mixed effect models.

The reviewer makes a similar comment below, so we respond to it there. But briefly, we agree and have now included this.

Line specific comments ==========

L27: There is already much information in the abstract, consider removing it 

We agree and have removed it.

L28: machine-learning, consistency! 

We’ve fixed this, and have also checked it throughout the manuscript.

L39: I would like to see a stronger conclusion here

We have not made this change, as we prefer to be more conservative given some of our low R2 values, and we feel that the sentence, as stated, accurately represents our findings.

L39: I am not exactly sure what you mean here with 'explanatory power'? The predictions of ml is a weak effect in the final predictive model for the indirect interactions?

This line now reads (L40): “Further, although our machine-learning informed methods could significantly predict indirect effects, the explanatory power of our machine-learning models for indirect interactions was reasonably low.”

L130-131: Yes, see also Poisot et al 2015 (10.1111/oik.01719) 

We’ve added this citation here.

L135: See Poisot et al 2015, you need distinguish yourself from this work (which is no problem but you have to show that you are well informed about the relevant literature)

Thank you for reminding us about this paper. We’ve now cited it in a couple of places. One of the main distinguishing features of our work is the indirect effect predictions, which we introduce later in the introduction.

L138: in the first line of the introduction you write 'nontarget' and here 'non-target', consistency! 

Thank you for pointing this out. We’ve now changed everything to ‘non-target’ throughout the manuscript.

L143-144: I assume that this is indeed possible, but to identify the (predictive) underlying rules which generalizes well over scales/habitat types would require many different datasets from different scales to control for all the scale effects and force the ML model to identify 'global' patterns.

Yes, we agree and discuss some of the difficulties of predicting networks across habitats in the Discussion in the section ‘Predicting networks within vs. across habitat types’.

L272: which feature importance? Gini?

Yes, we use Gini and we have added a few words on this line to say this.

L277: interesting, previous works reported that phylogenetic predictors are important.

We mention the importance of phylogenies on L212, but since this isn’t a main focus of our manuscript, we don’t discuss this point further. 

L281: I wonder if it wouldn't be better to correct the interaction outcomes by the abundances to force the models to learn other predictive rules. I fully agree with you that abundance can be a very important predictor but this is exactly the problem, because the model could mainly use abundance and other correlative predictors could be neglected, leading to a less generalisable model if in the next dataset the abundance pattern is different.

Please report the individual importances

We report the individual importances in the appendix. We have added ‘(Table 20 in Appendix S1)’ to L323. We responded to the rest of this comment above, where the reviewer asked a similar question. 

L346: I found this section difficult to follow, it might help the reader to give information about the regression models with the commonly known formula syntax (R, lme4)

We have included all the R formulas as a table in Appendix S1. We chose not to include the R formulas in the main text, as we felt this would involve some repetition (as we already explain in words the response variables, and each of the predictor variables and random effects). 

L370-375: Excellent!

Thanks!

L384: Why? 

This relates to a similar comment (about model selection) above, so we responded to it above. 

L423: Again, why? We also responded to this comment about model selection above. 

L457: Consistency? Why use previously AIC and now AICc? Could you please also provide more information about the dimensions of the data? E.g. how many observations did you have for the different habitat types?

We used AICc here because it is more suited to smaller sample sizes than AIC (as we mention on L527). On L517 we give the total number of observations. Because we did not include forest type as a fixed effect in these models (as explained in “Can we predict indirect effects using machine-learning (random forest and KNN) informed methods?” in the main text. These line numbers are for the model the reviewer’s comment was based on), we only include the total number of observations (N=20), rather than the number of observations for each habitat type (native forest = 11, plantation forest = 9). 

L523-525: Here, you interpreted the model causally, but you selected for the best predictive model via AIC 

The reviewer makes a similar comment above, so we responded to it above to avoid repetition. 

L581-584: I agree, finding a correct threshold for new data is very difficult/non-trivial

References

Bartomeus, I., D. Gravel, J. M. Tylianakis, M. A. Aizen, I. A. Dickie, and M. Bernard-Verdier. 2016. A common framework for identifying linkage rules across different types of interactions. Functional Ecology 30:1894-1903.

Canard, E. F., N. Mouquet, D. Mouillot, M. Stanko, D. Miklisova, and D. Gravel. 2014. Empirical Evaluation of Neutral Interactions in Host-Parasite Networks. The American Naturalist 183:468-479.

Eklöf, A., U. Jacob, J. Kopp, J. Bosch, R. Castro-Urgal, N. P. Chacoff, B. Dalsgaard, C. de Sassi, M. Galetti, P. R. Guimaraes, S. B. Lomascolo, A. M. Martin Gonzalez, M. A. Pizo, R. Rader, A. Rodrigo, J. M. Tylianakis, D. P. Vazquez, and S. Allesina. 2013. The dimensionality of ecological networks. Ecology Letters 16:577-583.

Frost, C. M., G. Peralta, T. A. Rand, R. K. Didham, A. Varsani, and J. M. Tylianakis. 2016. Apparent competition drives community-wide parasitism rates and changes in host abundance across ecosystem boundaries. Nature Communications 7:12644.

Holt, R. D. 1977. Predation, apparent competition, and the structure of prey communities. Theoretical Population Biology 12:197-229.

Jordano, P., J. Bascompte, and J. M. Olesen. 2003. Invariant properties in coevolutionary networks of plant–animal interactions. Ecology Letters 6:69-81.

Krishna, A., P. R. Guimarães, P. Jordano, and J. Bascompte. 2008. A Neutral-Niche Theory of Nestedness in Mutualistic Networks. Oikos 117:1609-1618.

Müller, C. B., I. C. T. Adriaanse, R. Belshaw, and H. C. J. Godfray. 1999. The structure of an aphid-parasitoid community. Journal of Animal Ecology 68:346-370.

Stang, M., P. G. Klinkhamer, and E. van der Meijden. 2007. Asymmetric specialization and extinction risk in plant–flower visitor webs: a matter of morphology or abundance? Oecologia 151:442-453.

Vázquez, D. P., N. P. Chacoff, and L. Cagnolo. 2009. Evaluating Multiple Determinants of the Structure of Plant-Animal Mutualistic Networks. Ecology 90:2039-2046.

Volkov, I., J. R. Banavar, S. P. Hubbell, and A. Maritan. 2003. Neutral theory and relative species abundance in ecology. Nature 424:1035-1037.

---

## [Decision Letter · Decision Letter 1]

6 Apr 2021

PONE-D-20-35961R1

Predicting direct and indirect non-target impacts of biocontrol agents using machine-learning approaches

PLOS ONE

Dear Dr. Kotula,

Thank you for submitting your manuscript to PLOS ONE. After careful consideration, we feel that it has merit but does not fully meet PLOS ONE’s publication criteria as it currently stands. Therefore, we invite you to submit a revised version of the manuscript that addresses the points raised during the review process.

You have done an excellent job responding to the first round of reviews.  However, both reviewers found some minor issues that still need to be addressed.  

There also may be some issue with the line numbers, the ones I was seeing in my draft obviously differed from those that reviewer 3 was looking at.  For clarity, I am reiterating reviewer 3's concerns: 

Line 357, which refers to equation one.  K does not occur anywhere in equation 1. 

Line 1007: I am not entirely sure what "error caused by gremlins"  reviewer three is referring to.  However, the journal capitalization on line 1007 is different from that of the rest of the references (only the first word is capitalized).  In reference 83, there is also some unnecessary capitalization of the article title.

It would be a good idea to double check the formatting of the references before your final submission, it’s very easy for minor errors to creep into the sources cited. 

As long as you can deal with the minor remaining issues raised by the reviewers, this should be ready to go.  I leave it to your discretion whether to implement reviewer three's other suggestion for figure 1.  It would enhance readability, but the figure seems fine in it's current state to me (and a great improvement to this manuscript!).

We look forward to receiving your revised manuscript.

Kind regards,

Patrick R Stephens, Ph.D.

Academic Editor

PLOS ONE

Journal Requirements:

Reviewers' comments:

Reviewer's Responses to Questions

**Comments to the Author**

1. If the authors have adequately addressed your comments raised in a previous round of review and you feel that this manuscript is now acceptable for publication, you may indicate that here to bypass the “Comments to the Author” section, enter your conflict of interest statement in the “Confidential to Editor” section, and submit your "Accept" recommendation.

Reviewer #2: All comments have been addressed

Reviewer #3: (No Response)

2. Is the manuscript technically sound, and do the data support the conclusions?

Reviewer #2: Yes

Reviewer #3: Yes

3. Has the statistical analysis been performed appropriately and rigorously? 

Reviewer #2: Yes

Reviewer #3: Yes

4. Have the authors made all data underlying the findings in their manuscript fully available?

Reviewer #2: Yes

Reviewer #3: Yes

5. Is the manuscript presented in an intelligible fashion and written in standard English?

Reviewer #2: Yes

Reviewer #3: Yes

6. Review Comments to the Author

Reviewer #2: Dear Dr. Stephens, dear authors,

This is my second review of the manuscript ‘predicting direct and indirect non-targets of biocontrol agents using machine-learning approaches’ by Kotula et al. and it was a pleasure to review the manuscript again

The authors have addressed and fulfilled all the points raised by both of the reviewers: they added a conceptional figure explaining their approach, they used a blocked CV, they fitted the ML models without using the species abundances as predictors, and they admitted to make all code/methods freely available.

I think that the manuscript in its current form would make a great contribution to the community and fits perfectly into the scope of PLOS ONE.

I added below a few general and specific comments.

===== General comments =====

Check for consistency, sometimes the authors write ‘random forest’ and then ‘random-forest’.

The authors said that they have now included the formulas for their mixed-effect models but I could not find them.

Optional: I think that the authors could make explicit connections between their research and its relevance for conservation and biodiversity ecology. For instance, the authors could expand their conclusion in the abstract which falls short compared to the rest of the abstract.

===== Specific comments =====

L258: either you provide some references for the Netflix problem (e.g. https://peerj.com/articles/3644/) or you omit this sentence

L306-308: Can you provide a reference for this statement?

L404-405: scaling doesn’t result in a sd of 1?

L434: I don’t think this avoids the problem of stepwise selection of variables because the selection after the AIC is itself the problem (the problem is that the AIC selects the best predictive model and not the causal one (BIC does, but only approximatively)). Maybe just omit the part ‘thereby avoiding…’

L702-717: Well, kNN is also a very ‘simple’ (less complex) ML algorithm compared to RF and kNN is also unable to infer automatically interactions between predictors and is less able to handle non-linearities

Reviewer #3: The authors present a very convincing and appealing case for the use of ML to infer potential consequences of biocontrol agents looking at both direct and indirect effects, in addition to comparing two different ML approaches. After the initial round of review it seems that both reviewers were primarily concerned with the ‘density’ of the manuscript and (barring some smaller comments) were very convinced by the presentation of the statistical/analysis portion. I believe that the authors have taken the feedback from the initial reviewers to heart and that the inclusion of the conceptual figure (Fig. 1) as well as some of the rephrasing of sections has made the manuscript substantially easier to follow and will surely make it more accessible to a general audience. I think showcasing the two streams of direct and indirect effects and how/when data are shared in the conceptual figure has made the storyline much easier to follow.

I would like to commend the authors on managing to condense all this information to be contained within a single manuscript and I have one stylistic comment/query regarding the conceptual figure and one point with regards to equation 1 (l. 366) but otherwise believe that the manuscript complies to the PLOS one guidelines and should be considered ready for publication.

I wonder if it may be possible to add short subheadings along with the letters to make the figure more intuitive at face value without having to refer to the figure legend? You already have this for a few of the points _e.g._ g, e, i and it would be a case of bolding them and ‘marrying’ them to the letter. This feeling might be in part due to the submission requirements of the journal and having the legend split form the figure may have contributed to me feeling the need for some subheading in the figure. I do just want to stress that I don't think altering the figure is _critical_ run order for the manuscript to progress.

Could the authors have a second look at the descriptive breakdown of equation 1? In line 266 the authors state that _k_ is a parasitoid species. However, there is no variable _k_ indicated in the actual equation.

## Minor comment

A gremlin crept into your reference list and tinkered with reference number 83 - line 1107

7. PLOS authors have the option to publish the peer review history of their article (what does this mean?). If published, this will include your full peer review and any attached files.

Reviewer #2: No

Reviewer #3: No

---

## [Author Response · Author response to Decision Letter 1]

10 May 2021

Kotula et al. Response to Reviewers

10 May 2021

Dear Dr. Stephens,

On behalf of my co-authors, I would like to thank you and the two reviewers for the helpful comments on our manuscript. My co-authors and I appreciate the suggestions, and we present point-by-point responses (in blue text) to each comment below, along with tracked changes in the submitted manuscript and supporting information. This is the second review of our manuscript, and we feel that the reviewer’s comments have all been suitably addressed, and that the manuscript is improved. Below we also provide an updated funding statement. If you have any further questions, concerns, or revisions, please do not hesitate to contact us.

Kind regards,

Hannah Kotula, Guadalupe Peralta, Carol Frost, Jacqui Todd, and Jason Tylianakis

Amended funding statement:

H.J.K acknowledges support from a Roland Stead Postgraduate Scholarship in Biology. HJK, GP and JMT were funded by the Marsden Fund (grant number UOC1705), administered by the Royal Society of New Zealand. J.H.T is funded by the Better Border Biosecurity (B3) (www.b3nz.org) research collaboration funded by the New Zealand Government.

We acknowledge that one of the authors (J.H.T) is employed by a commercial company (The New Zealand Institute for Plant and Food Research Limited). Although this company does both government-funded and industry-funded work, there is no industry funding for this research. The funder provided support in the form of salaries for an author [J.H.T], but did not have any additional role in the study design, data collection and analysis, decision to publish, or preparation of the manuscript. The specific role of this author is articulated in the ‘author contributions’ section.

Editor’s comments

Thank you for submitting your manuscript to PLOS ONE. After careful consideration, we feel that it has merit but does not fully meet PLOS ONE’s publication criteria as it currently stands. Therefore, we invite you to submit a revised version of the manuscript that addresses the points raised during the review process.

You have done an excellent job responding to the first round of reviews. However, both reviewers found some minor issues that still need to be addressed.

There also may be some issue with the line numbers, the ones I was seeing in my draft obviously differed from those that reviewer 3 was looking at. For clarity, I am reiterating reviewer 3's concerns:

Thank you for your comments, and we apologise for the difference in line numbers. Thanks also for reiterating reviewer 3’s comments.

Line 357, which refers to equation one. K does not occur anywhere in equation 1.

Thanks for noticing that k doesn’t occur in equation 1, and we apologise for this mistake. We have now removed “k is a parasitoid species” from line 358.

Line 1007: I am not entirely sure what "error caused by gremlins" reviewer three is referring to. However, the journal capitalization on line 1007 is different from that of the rest of the references (only the first word is capitalized). In reference 83, there is also some unnecessary capitalization of the article title.

It would be a good idea to double check the formatting of the references before your final submission, it’s very easy for minor errors to creep into the sources cited.

Thanks for pointing this out. We have now double checked the formatting of the references for both the main text and the appendix. 

As long as you can deal with the minor remaining issues raised by the reviewers, this should be ready to go. I leave it to your discretion whether to implement reviewer three's other suggestion for figure 1. It would enhance readability, but the figure seems fine in its current state to me (and a great improvement to this manuscript!).

Thanks for this. We agree and have kept the figure as it was (see response to reviewer 3’s suggestion for figure 1 below). 

Reviewer #2: Dear Dr. Stephens, dear authors,

This is my second review of the manuscript ‘predicting direct and indirect non-targets of biocontrol agents using machine-learning approaches’ by Kotula et al. and it was a pleasure to review the manuscript again

The authors have addressed and fulfilled all the points raised by both of the reviewers: they added a conceptional figure explaining their approach, they used a blocked CV, they fitted the ML models without using the species abundances as predictors, and they admitted to make all code/methods freely available.

I think that the manuscript in its current form would make a great contribution to the community and fits perfectly into the scope of PLOS ONE.

I added below a few general and specific comments.

Thanks for your comments. Yes, our code is now available on GitHub (https://github.com/hannah-jk/predicting_nontarget_impacts), and we have now included this link in the main text (just below the acknowledgements section). 

===== General comments =====

Check for consistency, sometimes the authors write ‘random forest’ and then ‘random-forest’.

Thanks for pointing this out. We have now double checked this and we now always use “random forest” (except in cases where it is used as a compound adjective, like “random-forest model” where we use a hyphen). 

The authors said that they have now included the formulas for their mixed-effect models but I could not find them.

We have included the formulas for the mixed-effect models in Appendix S1 (Tables 21-24). We have now added “The R formulas for these, and all subsequent, models are included in Appendix S1 (Tables 21-24)” to line 414 to make this clear.

Optional: I think that the authors could make explicit connections between their research and its relevance for conservation and biodiversity ecology. For instance, the authors could expand their conclusion in the abstract which falls short compared to the rest of the abstract.

Thanks for the suggestion. The last sentence of the abstract now reads:

“Combining machine-learning and network approaches provides a starting point for reducing risk in biocontrol introductions, and could be applied more generally to predicting species interactions such as impacts of invasive species.”

===== Specific comments =====

L258: either you provide some references for the Netflix problem (e.g. https://peerj.com/articles/3644/) or you omit this sentence

We have now added the reference you suggest. 

L306-308: Can you provide a reference for this statement?

Yes, we have now added a reference for this statement. 

L404-405: scaling doesn’t result in a sd of 1?

We used the scale function from base R with centre=TRUE, scale=TRUE which scales variables so that they have a mean of zero and a standard deviation of one. We have added a few words to this sentence in the manuscript to make it more clear, and it now reads (L404-407):

“For each model, we scaled and centred the fixed effect (by subtracting the mean and dividing by the standard deviation such that the scaled variable would have a mean of zero and a standard deviation of one) to help with model convergence.”

L434: I don’t think this avoids the problem of stepwise selection of variables because the selection after the AIC is itself the problem (the problem is that the AIC selects the best predictive model and not the causal one (BIC does, but only approximatively)). Maybe just omit the part ‘thereby avoiding…’

We agree and we have now omitted “thereby avoiding issues associated with stepwise selection of variables.”

L702-717: Well, kNN is also a very ‘simple’ (less complex) ML algorithm compared to RF and kNN is also unable to infer automatically interactions between predictors and is less able to handle non-linearities

Thanks for this comment. We agree and we have now added (L719-722): “Additionally, random forest is a more complex algorithm than KNN. Because random forest is based on many trees (each with many branches), random forest is able to account for interactions between variables and non-linearities. This may also have contributed to the better predictive ability of random forest compared to KNN.”

Reviewer #3: The authors present a very convincing and appealing case for the use of ML to infer potential consequences of biocontrol agents looking at both direct and indirect effects, in addition to comparing two different ML approaches. After the initial round of review it seems that both reviewers were primarily concerned with the ‘density’ of the manuscript and (barring some smaller comments) were very convinced by the presentation of the statistical/analysis portion. I believe that the authors have taken the feedback from the initial reviewers to heart and that the inclusion of the conceptual figure (Fig. 1) as well as some of the rephrasing of sections has made the manuscript substantially easier to follow and will surely make it more accessible to a general audience. I think showcasing the two streams of direct and indirect effects and how/when data are shared in the conceptual figure has made the storyline much easier to follow.

I would like to commend the authors on managing to condense all this information to be contained within a single manuscript and I have one stylistic comment/query regarding the conceptual figure and one point with regards to equation 1 (l. 366) but otherwise believe that the manuscript complies to the PLOS one guidelines and should be considered ready for publication.

Thanks for your comments. 

I wonder if it may be possible to add short subheadings along with the letters to make the figure more intuitive at face value without having to refer to the figure legend? You already have this for a few of the points _e.g._ g, e, i and it would be a case of bolding them and ‘marrying’ them to the letter. This feeling might be in part due to the submission requirements of the journal and having the legend split form the figure may have contributed to me feeling the need for some subheading in the figure. I do just want to stress that I don't think altering the figure is _critical_ run order for the manuscript to progress.

Thanks for your suggestion. We feel that the figure as it stands has about the right amount of text. When the figure and legend are together, we feel they clearly explain our methods. 

Could the authors have a second look at the descriptive breakdown of equation 1? In line 266 the authors state that _k_ is a parasitoid species. However, there is no variable _k_ indicated in the actual equation.

Thanks for noticing this, and we apologise for this mistake. The editor reiterated this comment above, and we have responded to it there.

## Minor comment

A gremlin crept into your reference list and tinkered with reference number 83 - line 1107

Thanks for noticing this. We have now double checked the reference list.

---

## [Editor Report · Decision Letter 2]

17 May 2021

Predicting direct and indirect non-target impacts of biocontrol agents using machine-learning approaches

PONE-D-20-35961R2

Dear Dr. Kotula,

We’re pleased to inform you that your manuscript has been judged scientifically suitable for publication and will be formally accepted for publication once it meets all outstanding technical requirements.

Kind regards,

Patrick R Stephens, Ph.D.

Academic Editor

PLOS ONE

---

## [Editor Report · Acceptance letter]

21 May 2021

PONE-D-20-35961R2 

Predicting direct and indirect non-target impacts of biocontrol agents using machine-learning approaches 

Dear Dr. Kotula:

I'm pleased to inform you that your manuscript has been deemed suitable for publication in PLOS ONE. Congratulations! Your manuscript is now with our production department. 

Kind regards, 

on behalf of

Dr. Patrick R Stephens 

Academic Editor

PLOS ONE